# Heterogeneity in strategy use during arbitration between experiential and observational learning

Caroline J. Charpentier [1,2] ✉, Qianying Wu [1], Seokyoung Min[1], Weilun Ding[1], Jeffrey Cockburn [1] & John P. O'Doherty [1]

To navigate our complex social world, it is crucial to deploy multiple learning strategies, such as learning from directly experiencing action outcomes or from observing other people's behavior. Despite the prevalence of experiential and observational learning in humans and other social animals, it remains unclear how people favor one strategy over the other depending on the environment, and how individuals vary in their strategy use. Here, we describe an arbitration mechanism in which the prediction errors associated with each learning strategy influence their weight over behavior. We designed an online behavioral task to test our computational model, and found that while a substantial proportion of participants relied on the proposed arbitration mechanism, there was some meaningful heterogeneity in how people solved this task. Four other groups were identified: those who used a fixed mixture between the two strategies, those who relied on a single strategy and non-learners with irrelevant strategies. Furthermore, groups were found to differ on key behavioral signatures, and on transdiagnostic symptom dimensions, in particular autism traits and anxiety. Together, these results demonstrate how large heterogeneous datasets and computational methods can be leveraged to better characterize individual differences.

As humans, we learn about the world around us by seeking and integrating information from multiple sources. On the one hand, we heavily rely on our own past experience to predict the future. Experiential learning (EL) is such that actions that were rewarded in the past tend to be repeated, while actions that were punished in the past tend to be avoided. EL can be relied on to solve many reinforcement learning problems, from learning simple associations between stimulus, action and reward (model-free learning) to complex cognitive maps (model-based learning) and exploitation/exploration trade-offs[1–4]. On the other hand, as a social species with sophisticated social skills that allow us to make collective decisions and function in society, humans can learn from observing others[5–7]. Such observational learning (OL) is thought to confer the evolutionary advantageous ability to assess the consequences of actions available in the environment without having to directly experience the potentially negative outcomes of those actions. OL is prevalent across many domains, from basic sensory-motor learning[8–10] to complex strategic decision-making[11,12], from aversive[13,14] to reward learning[15–18], and can even extend to learning from non-human agents[19] or from replayed actions[18].

Depending on the uncertainty of the environment, a given strategy may become more reliable to deploy at different points in times[2], consistent with a "mixture of experts" framework in which different expert systems take the lead in guiding behavior when their predictions are most reliable[20]. Evidence for such uncertainty- or reliability-based arbitration between learning strategies, as well as its neural correlates, has been provided within each domain. In EL, people dynamically arbitrate between model-free and model-based

[1]Division of Humanities and Social Sciences, California Institute of Technology, Pasadena, CA, USA. [2]Department of Psychology & Brain and Behavior Institute, University of Maryland, College Park, MD, USA. ✉e-mail: ccharpen@umd.edu

learning[21,22]. In OL, recent evidence suggests a similar arbitration mechanism between imitation – the tendency to repeat other people's choices – and emulation – the tendency to infer their goals[18], as well as between cooperative and competitive learning during strategic interactions[23]. Yet, whether and how people may engage dynamic arbitration processes across domains – that is, between EL and OL – remains unclear. How people integrate experiential and social information during learning has been the focus of a lot of research in the past two decades. Multiple studies have shown that not only are decisions influenced by both sources of information[24,25], but experiential and social learning signals co-exist in the brain and can be integrated to predict decisions[15,26–33]. However, these studies have not directly assessed the possibility of a dynamic arbitration mechanism between the two learning domains. In other words, it remains unclear whether the weight attributed to each strategy before making a decision varies depending on the environment.

Here, we designed and optimized a novel task that probes both EL and OL and manipulates uncertainty in each strategy's predictions to promote dynamic arbitration. We collected data in two independent online studies, to test the predictions that when outcomes are more predictable, EL should be favored, and when inferences from the observed agent's actions are more reliable, OL should be favored. A second goal of this study was to characterize heterogeneity in the strategies participants deploy during learning. Given the recent explosion (partly driven by COVID-19) of online data collection, it has become clear that, despite attention checks, performance in online studies tends to be noisier than in the lab, most likely because of the uncontrolled environment, lack of direct interaction between participant and experimenter, and larger possible distractions[34,35]. However, online studies allow for the collection of large-scale datasets in shorter timeframes, often exhibiting good replicability of in-lab findings[36], thus providing increased power for a more thorough characterization of individual differences and of their relevance to psychopathology. Therefore, in addition to probing the dynamic arbitration framework described above, we also investigated the possibility that not all participants relied on this computational model to solve the task[37]. Though such heterogeneity is likely to exist in any study sample (online and in-person studies, clinical and general populations, etc), it is not usually well characterized in existing studies, given that sample sizes are too small or that most computational modelling approaches tend to select a "winning" model and apply it to all participants. Here, we predicted that different groups of people might rely on different strategies and set out to characterize this heterogeneity. Specifically, we tested for the possibility that in addition to dynamic arbitration, some individuals might combine the two strategies in a less flexible way, such as by relying on an unchanging allocation between the two strategies (without dynamically arbitrating), or that some might instead predominantly rely on a single strategy (either EL-only; or OL-only) to solve the task, while others might use irrelevant heuristics, such as preferring a given action (left versus right) or a given stimuli throughout. We also tested whether groups that are solely defined based on model-fitting would differ from each other in meaningful ways in their behavior on the task and in transdiagnostic symptom dimensions.

Recent literature in computational psychiatry has shown that anxiety is associated with difficulties in adapting to volatility and changes in uncertainty[38–40], increased exploration to reduce uncertainty[41], and faster learning from negative outcomes[42,43]. Social anxiety has also been found to be associated with excessive deliberation[44] and with suboptimal learning[45]. Finally, autism has been linked to deficits in behavioral adaptation during social inference[46], specifically suboptimal flexibility and lower mentalizing sophistication[47], overestimation of the volatility of sensory environment[48], reduced implicit causal inference about sensory signals[49], and enhanced observational learning in the aversive domain[50]. Therefore, we hypothesized that individual differences in subclinical traits related to anxiety, autism and social anxiety are likely to be sensitive to the computational heterogeneity in strategy use during EL, OL, and the arbitration between them.

In this work, we show that there is substantial heterogeneity in how participants solve this task, and that individuals can be reliably characterized by the computational model that best explains their behavior. We additionally validate this heterogeneity by demonstrating marked differences in key behavioral markers across groups, as well as differences in subclinical transdiagnostic traits related to autism and anxiety.

## Results

### Behavioral evidence for learning and mixture of strategies

Two groups of participants (Study 1: $N = 126$, Study 2: $N = 493$, see Methods for details) performed a novel task online designed to separately quantify experiential and observational learning tendencies during behavior (Fig. 1). In the task (160 trials), participants learn which of two tokens (orange or blue) is more likely to yield a reward, which can be achieved by observing another player choose between two boxes (identified by unique fractals superimposed on each box) to obtain a token, or through direct experience of the outcome associated with the chosen token (Fig. 1A). Importantly, participants were instructed that the other player knew which token was more valuable at any point in time and were instead learning which of the two boxes was more likely to yield the valuable token. By observing the other player's choices, one can thus infer which token color they were targeting as having the highest value. To promote continuous learning, as well as push the balance between EL and OL and test our proposed uncertainty-based arbitration mechanism, we manipulated the token reward probability (including reversals as well as periods of low vs high uncertainty) and the box-to-token transition probability (also alternating between periods of low and high uncertainty), depicted by blue and orange lines, respectively, in Fig. 1B. We also manipulated the variance in the reward magnitude so that in some blocks, when a token was rewarded, the variance in magnitude was high, and in other blocks the variance was low (see Methods for details). While this did not directly affect the reliability of EL predictions – that is, the ability to predict the occurrence of a reward remained the same – we hypothesized that high variance may constitute a form of (task-irrelevant) uncertainty and tested whether it played a role in the arbitration process, whereby EL may be weighed less in the high variance condition.

We first examined mean behavioral accuracy (probability of choosing the more valuable token, calculated across all trials). Accuracy was 0.582 ($\pm 0.087$ SD) in Study 1 and 0.604 ($\pm 0.083$ SD) in Study 2, significantly above chance level of 0.5 (Study 1: t(125) = 10.67, $P < 0.001$, $d = 0.94$, 95% CI [0.067, 0.097]; Study 2: t(492) = 26.08, $P < 0.001$, $d = 1.25$, 95% CI [0.097, 0.111]). Behavioral evidence for learning behavior was then obtained by calculating trial-by-trial accuracy for the first 8 trials after a reversal in token values. There was a clear increase in accuracy throughout those 8 trials, from 0.528 directly after a reversal to 0.60 in Study 1 and from 0.544 to 0.619 in Study 2. This increase, modelled as a linear main effect of trial in a mixed-effect linear model predicting accuracy (lme4 package in R, including a random intercept, followed by Type III ANOVA), was statistically significant (Study 1: F(1,875) = 33.03, $P < 0.001$, $\eta_p^2 = 0.036$, 95% CI [0.0067, 0.014], Fig. 2A; Study 2: F(1,3423) = 96.14, $P < 0.001$, $\eta_p^2 = 0.027$, 95% CI [0.007, 0.011], Fig. 2B).

We then classified whether participants' choice on each trial was consistent with experiential learning and with observational learning (see Fig. 1C, D for an illustration). Out of the trials where the two strategies predicted different choices according to this classification, we then calculated the proportion of choices consistent with OL (vs EL) as an index of preference for one or the other strategy. Mean OL choice propensity was 0.515 ($\pm 0.095$) in Study 1 (Fig. 2C) and 0.493 ($\pm 0.107$)

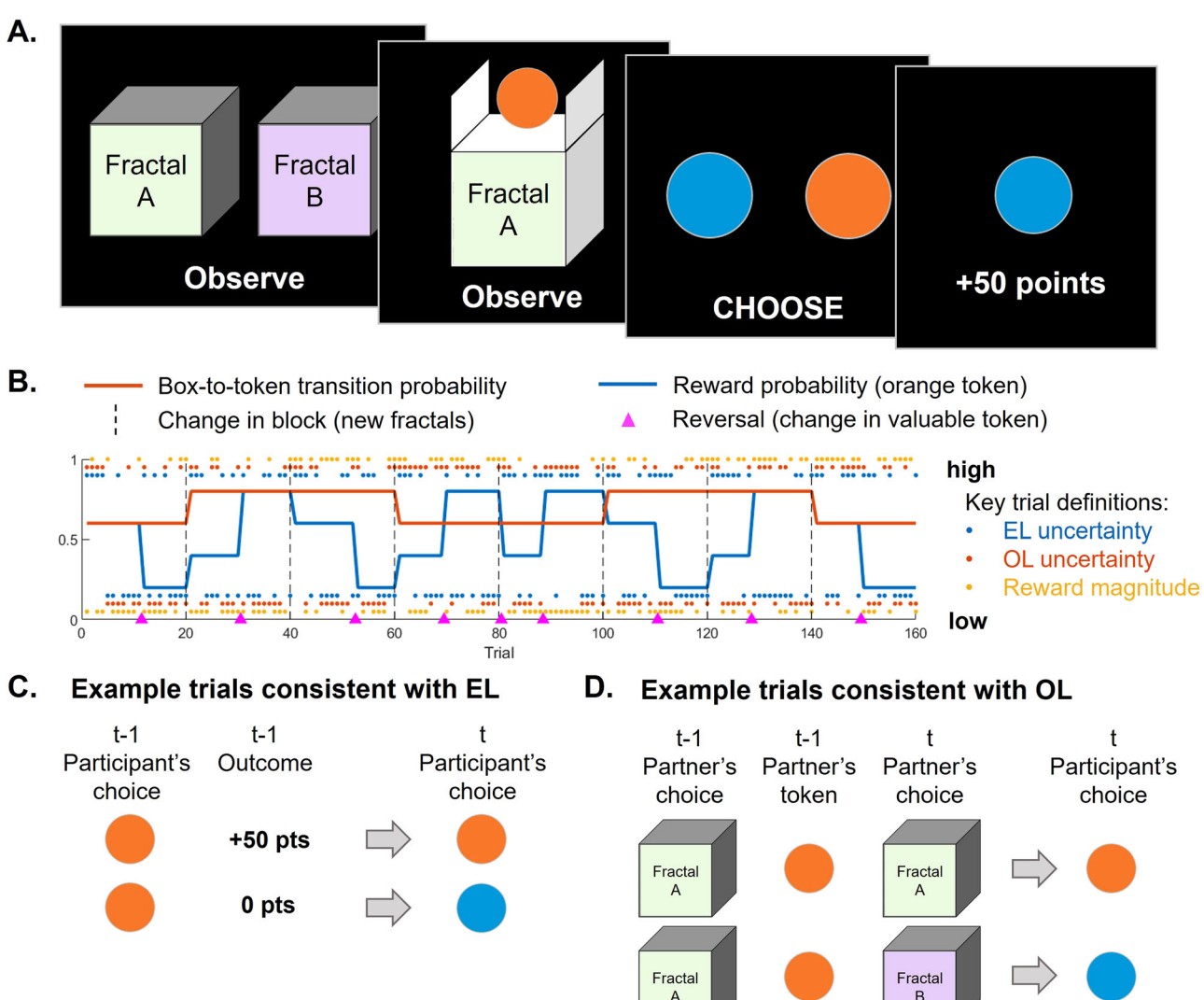

**Fig. 1 | Observational learning (OL) & Experiential learning (EL) task design.**
**A** On each trial, participants first observe another agent choose between two boxes represented by fractal images, then observe which token was obtained by the agent from the chosen box. Then participants choose for themselves between the two tokens and receive an outcome (from 0 – no reward – to 99 points) associated with the chosen token. From the 'observe' part of the trial, participants can learn from observation which token the other agent is trying to get. From the 'play' part of the trial, participants can learn from directly experiencing the outcomes associated with each token. **B** Example time course of probabilities, condition and block changes, and reversals. The task contained 8 blocks of 20 trials each. Each block started with a new pair of boxes (fractals), which had a transition probability towards their corresponding token of either 0.8 or 0.6, depicted by the orange line.

Within each block there was one reversal in the valuable token, depicted by the magenta triangles, with the blue line representing the reward probability associated with the orange token ( = 1 – P(reward | blue token)). While the lines represent EL and OL uncertainty conditions, for behavioral analyses, we defined key uncertainty trials as follows: EL uncertainty (blue points) was deemed low on trials where past outcome-action-outcome sequence was consistent, and high otherwise. OL uncertainty (orange points) was deemed low if the past two box-token transitions were consistent, and high otherwise. Finally, reward magnitude was considered low if the past outcome magnitude was equal to or below 25 points, and high otherwise. **C, D** Illustration of the trial definitions used to classify trials as consistent with EL (**C**) and consistent with OL (**D**).

in Study 2 (Fig. 2D), not significantly different from 0.5 (both $d < 0.16$, $P > 0.08$, Study 1: $BF_{10} = 0.436$, Study 2: $BF_{10} = 0.163$). This implies that both OL and EL strategies were relied on approximately equally across participants and across studies, although there was also substantial individual variability in the degree of engagement of these two strategies, with some participants exhibiting a clear preference for one strategy or the other.

To more formally assess whether participants used a mixture of the two strategies during the task, we ran a mixed-effects general linear model (ME-GLM) predicting choice on each trial from the outcome of the past trial (signature of EL) and from the partner's past choice (signature of OL). We found that both effects were significant, both in Study 1 (EL fixed effect: estimate = 0.357 ± 0.051 (SE), t(19862) = 6.94,

$P < 0.001$; OL fixed effect: estimate = 0.216 ± 0.026 (SE), t(19862) = 8.21, $P < 0.001$; Fig. 2E) and in Study 2 (EL fixed effect: estimate = 0.519 ± 0.025 (SE), t(77960) = 20.74, $P < 0.001$; OL fixed effect: estimate = 0.241 ± 0.012 (SE), t(77960) = 19.60, $P < 0.001$; Fig. 2F), indicative of hybrid behavior between OL and EL (see Table S1A for all statistics).

**Uncertainty-driven behavioral changes in strategy**
We next examined whether participants flexibly switched between OL and EL depending on the variations in uncertainty. First, we classified trials as low versus high OL uncertainty trials and low versus high EL uncertainty trials depending on the recent trial history (Fig. 1B, see Methods for details). Those trials broadly overlapped with the low vs

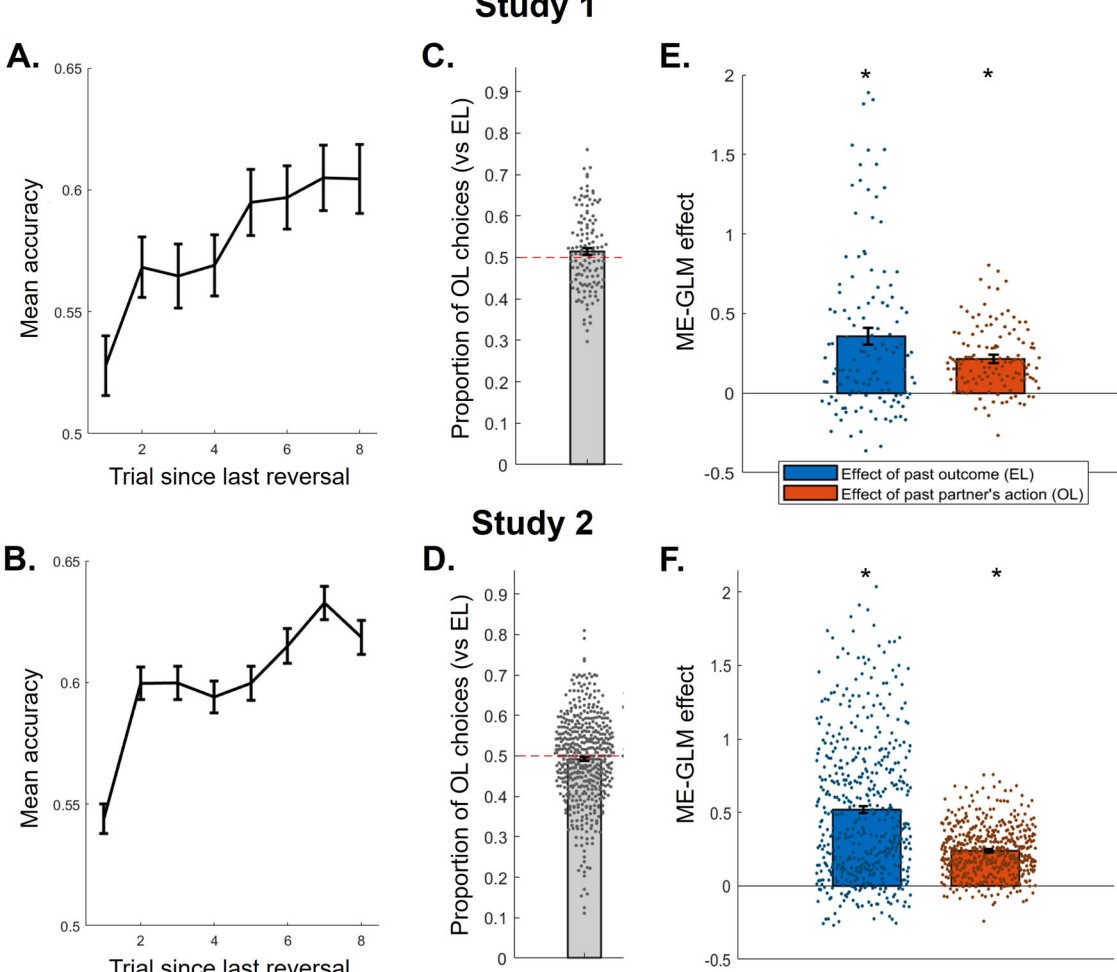

**Fig. 2 | Behavioral signatures of learning and strategy use. A, B** Behavioral evidence for learning behavior for Study 1 (**A**) and Study 2 (**B**), calculated as the mean accuracy (choice of correct token) for each of the first 8 trials following a reversal in token values, then averaged across participants. Error bars represent SEM. **C, D** The proportion of choices consistent with observational learning (OL) versus experiential learning (EL) was calculated out of the trials where OL and EL made different predictions (according to the definitions depicted in Fig. 1. **C, D** For Study 1 (**C**) and Study 2 (**D**). Each dot depicts an individual participant. **E, F** Main effects of past outcome (EL effect, blue) and of past partner's action (OL effect, orange) on current participant's choice were quantified in a mixed-effects generalized linear model (ME-GLM), for Study 1 (**E**) and Study 2 (**F**). Bars represent the fixed effect coefficient estimates; error bars represent the standard error associated with those estimates; stars represent the significance of the fixed effects obtained from the ME-GLM (two-sided, all $P < 0.001$); and each dot is an individual participant (random effect); Study 1: $N = 126$ independent participants (**A, C, E**); Study 2: $N = 493$ independent participants (**B, D, F**).

high uncertainty conditions that were defined by design (larger proportion of low OL uncertainty trials in low compared to high OL uncertainty conditions: Study 1: $t(125) = 30.3$, Study 2: $t(492) = 61.1$; larger proportion of low EL uncertainty trials in low compared to high EL uncertainty conditions: Study 1: $t(125) = 52.2$, Study 2: $t(492) = 91.5$; all $Ps < 0.001$), but were defined to capture trial-by-trial variations in uncertainty. We hypothesized those variations would be more representative of how dynamic changes in uncertainty were experienced by participants, given that actual changes in uncertainty were not cued, which would lead to a lag in information integration when considering the blocked conditions. Indeed, we found that uncertainty *trials* were stronger predictors of choice throughout the task than uncertainty *conditions* (Table S2). We then calculated the breakdown of OL choice propensity as defined above (illustrated in Fig. 1C, D, data shown in Fig. 2C, D) across these uncertainty trial types and tested their significance in a linear mixed-effects model predicting OL (vs EL) choice propensity for each participant from OL uncertainty trial type, EL uncertainty trial type, and their interaction. We found that the main effect of both factors was significant, both in Study 1 (effect of OL uncertainty trial type: $F(1,833) = 32.64$, $P < 0.001$, $\eta_p^2 = 0.038$, 95% CI

[0.014, 0.107]; effect of EL uncertainty trial type: $F(1,833) = 39.31$, $P < 0.001$, $\eta_p^2 = 0.045$, 95% CI [−0.186, −0.093]; Fig. 3A) and in Study 2 (effect of OL uncertainty trial type: $F(1,3234) = 149.88$, $P < 0.001$, $\eta_p^2 = 0.044$, 95% CI [0.056, 0.105]; effect of EL uncertainty trial type: $F(1,3234) = 268.27$, $P < 0.001$, $\eta_p^2 = 0.077$, 95% CI [−0.196, −0.148]; Fig. 3B). Moreover, there was also a significant interaction between EL and OL uncertainty trial types (Study 1: $F(1,833) = 4.31$, $P = 0.038$, $\eta_p^2 = 0.005$, 95% CI [0.0038, 0.135]; Study 2: $F(1,3234) = 9.896$, $P = 0.0017$, $\eta_p^2 = 0.003$, 95% CI [0.021, 0.090]), such that the effect of OL uncertainty was stronger when EL uncertainty was low. For comparison, the same analysis conducted on uncertainty *conditions* instead of *trials* is shown in Fig. S1.

Note that by design, and as explained above, we manipulated the variance in reward magnitude, with the prediction that high variance in reward magnitude may reduce the tendency to rely on EL, and therefore indirectly promote OL. However, in Study 1 we found that reward magnitude variance had no effect on OL vs EL choice propensity ($t(125) = 0.73$, $P = 0.46$, $d = 0.065$, $BF_{10} = 0.129$), which was also found to be the case in Study 2 ($t(492) = 1.68$, $P = 0.095$, $d = 0.076$, $BF_{10} = 0.203$). Instead, whether the magnitude itself was high or low

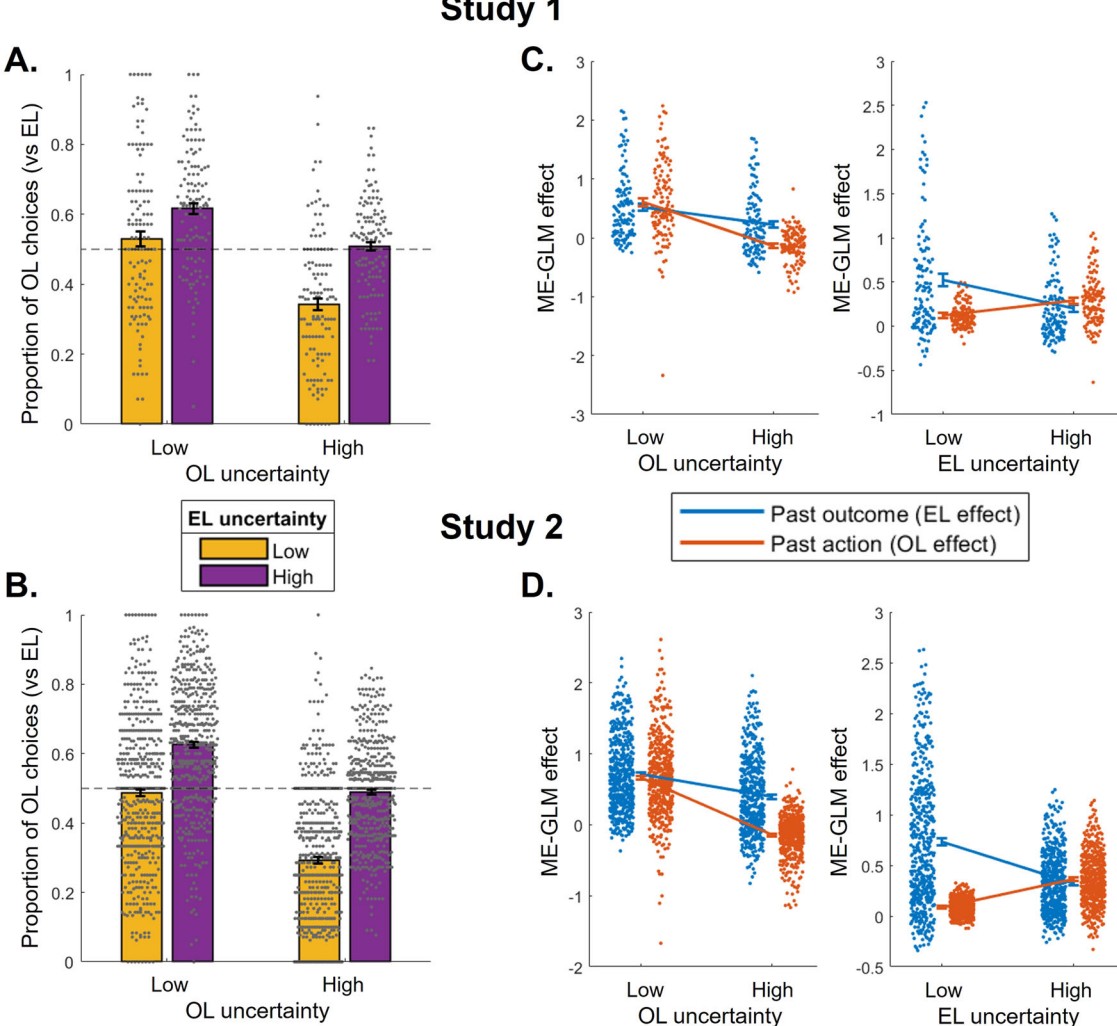

**Fig. 3 | Behavioral signature of uncertainty-driven arbitration between experiential (EL) and observational learning (OL). A, B** The proportion of OL choices was computed like in Fig. 2C,D, but separately for each of 4 trial types defined by OL uncertainty (low or high) and EL uncertainty (low or high), for Study 1 (**A**) and Study 2 (**B**). See Methods for details about how uncertainty trial types were defined, and Fig. 1B for an illustration. Each dot is an individual participant; error bars represent SEM. **C, D** Separate mixed-effects generalized linear models (ME-GLM) were run to quantify the effect of each uncertainty manipulation on EL (blue) and OL (orange) separately. Both fixed and random effects of past partner's action and past outcome, for high and low uncertainty trials, were included into the ME-GLM, allowing us to quantify each effect for low and high OL uncertainty trials (left), and low and high EL uncertainty trials (right), for Study 1 (**C**) and Study 2 (**D**). Data represent the fixed effect coefficient estimates for each uncertainty trial type; error bars represent the standard error associated with those estimates; and each dot is an individual participant (random effect); Study 1: $N = 126$ independent participants (**A, C**); Study 2: $N = 493$ independent participants (**B, D**). See Table S1 for statistics.

(defined as higher or lower than the mean expected reward of 25 points, see yellow dots on Fig. 1B), had a strong effect, with a lower propensity to rely on OL (therefore, higher propensity to rely on EL) when reward magnitude was high than low (Study 1: $t(125) = 6.08$, $P < 0.001$, $d = 0.54$; Study 2: $t(492) = 15.63$, $P < 0.001$, $d = 0.70$). Therefore, we used magnitude, rather than variance, as a condition to examine arbitration, but because those predictions were not part of our initial uncertainty-driven arbitration hypothesis, we opted to focus the analyses on the effects of OL and EL uncertainty trial types only in the main text, and we present the additional findings related to magnitude as a supplementary analysis in Fig. S2.

We then ran two separate ME-GLMs (one for each manipulation) to specifically quantify the effect of EL and OL uncertainty trial types on each strategy separately. In the previous analyses, we found effects of both manipulations on OL vs EL choice propensity in the expected direction, however, looking only at this behavioral metric we cannot disentangle whether, for example, OL uncertainty impacts behavior by increasing OL, decreasing EL, or both. With ME-GLMs quantifying both OL and EL effects we can address this. Each ME-GLM included four predictors of choice on each trial (both as fixed and random effects): past outcome for low and high uncertainty trials, and past partner action for low and high uncertainty trials (see Table S1B, C for statistics). The resulting random effects were then compared in a 2-by-2 ANOVA, revealing significant interactions between the strategy (OL vs EL effect) and the manipulation of interest, both in Study 1 (strategy * OL uncertainty: $F(1,375) = 31.06$, $P < 0.001$, $\eta_p^2 = 0.076$, 95% CI [0.284, 0.594]; strategy * EL uncertainty: $F(1,375) = 99.87$, $P < 0.001$, $\eta_p^2 = 0.21$, 95% CI [−0.574, −0.385]; Fig. 3C) and in Study 2 (strategy * OL uncertainty: $F(1,1467) = 176.00$, $P < 0.001$, $\eta_p^2 = 0.107$, 95% CI [0.409, 0.551]; strategy * EL uncertainty: $F(1,1467) = 752.21$, $P < 0.001$, $\eta_p^2 = 0.339$, 95% CI [−0.720, −0.624]; Fig. 3D). Crucially, the interactions were driven by a stronger effect of uncertainty trial type on the relevant strategy. In high OL uncertainty trials, the tendency to rely on OL was reduced more strongly than the tendency to rely on EL (Fig. 3C, D left). And interestingly, high EL uncertainty trials impacted the reliance on both strategies in opposite directions, that is, not only were they associated with a reduction in EL but also with an increase in OL (Fig. 3C, D right).

**Table 1 | Summary of model fits**

| Model | $N_{par}$ | Study 1 | | | | Study 2 | | | |
|---|---|---|---|---|---|---|---|---|---|
| | | AIC | OOS acc | Frequency | $N_{best}$ (%$_{tot}$) | AIC | OOS acc | Frequency | $N_{best}$ (%$_{tot}$) |
| Baseline | 4 | 215.7 | 0.521 | 0.205 | 25 (19.8) | 219.8 | 0.511 | 0.159 | 83 (16.8) |
| Experiential learning | 3 | 207.7 | 0.539 | 0.147 | 21 (16.7) | 204.9 | 0.552 | 0.060 | 24 (4.9) |
| Observational learning | 2 | 197.2 | 0.569 | 0.311 | 40 (31.8) | 194.8 | 0.575 | 0.190 | 95 (19.3) |
| Fixed mixture | 6 | 191.0 | 0.593 | 0.115 | 14 (11.1) | 186.0 | 0.607 | 0.324 | 160 (32.5) |
| Dynamic arbitration | 6 | 191.2 | 0.595 | 0.222 | 26 (20.6) | 187.1 | 0.608 | 0.267 | 131 (26.6) |

Each of the five models ($N_{par}$ = number of parameters) was fitted to participants data first using Matlab's *cbm* toolbox. Using individual model-fitting, we computed the mean AIC as well as mean out-of-sample accuracy (OOS acc) across participants. OOS accuracy was calculated for each individual by fitting the model on 7 task blocks and using the best-fitting parameters to calculate the likelihood of predicting the participant's choices in the remaining block (then iterating across all 8 blocks). We then used *cbm*'s hierarchical Bayesian inference fitting across all five models to compute model frequency. Selecting the best-fitting model for each individual participant (highest model responsibility), we then calculated the number and proportion of participants for whom each model explains their data best ($N_{best}$ column).

## Computational modelling: heterogeneity in strategy use

To assess whether participants' behavior was better explained by a single unitary model, or whether different individuals deploy different strategies, we fit a set of five parsimonious models to each participant's data. Specifically, the models included both single strategy models (EL only, OL only), a mixture model that combines the two strategies using a fixed weight, a dynamic arbitration model in which the weight varies dynamically depending on the reliability of each strategy, and finally, a baseline model that captures irrelevant, non-leaning strategies (see Methods for details and equations). We first tested whether the models were uniquely identifiable by calculating a confusion matrix (Fig. S3A). This analysis showed that all five models can be perfectly separated from each other, such that data generated by any given model is best explained by that model (exceedance probability of 1 relative to all the other models). Parameter recovery analyses were also performed for each model, consistently showing high correlations between actual and recovered parameters (Fig. S3B–F).

Model fitting to data was performed using hierarchical Bayesian inference in Matlab's *cbm* toolbox[51], both for each model separately to ensure reliable parameter estimates and including all five models as a set for Bayesian model comparison. Model frequencies from the latter analysis, as well as AIC and out-of-sample model predictive accuracy averaged across participants (see Methods for details) are reported in Table 1. Overall, those findings suggest that there was no clear and consistent winner. In both studies, the AIC values suggest a marginal advantage for the fixed mixture model, while out-of-sample accuracy slightly favored the dynamic arbitration model. Additionally, the model frequency values suggest somewhat of an even split, with no model exhibiting a frequency higher than 33%, with Study 1 showing the largest frequency for the observational learning model (31.1%) and Study 2 for the fixed mixture model (32.4%). Therefore, we reasoned that not every participant's data may be best explained by a single model across the group as a whole, and that instead, the data may be better analyzed by taking into account the best-fitting model for each participant. To do that, we relied on the individual model frequency values (model responsibility values provided as an output of *cbm* hierarchical Bayesian inference) to classify participants into five groups based on each participant's highest responsibility value. Group sizes are provided in Table 1, consistent with our hypothesized heterogeneity in strategy use.

## Posterior predictive checks

Posterior predictive checks were performed on the models using participants' best-fitting parameters. We first demonstrated the clear dissociation between the EL and OL models, showing that each model generates choices consistent with its predictions, and that our behavioral signature of interest was recovered by each model as expected depending on participants' preferred strategy (Fig. S4). Through more in-depth simulations, we then proceeded to generate data from each

model using participants' best-fitting parameters, and ran the mixed-effects GLMs shown in Fig. 2E, F (signature of hybrid EL/OL behavior) and in Fig. 3C, D (effect of EL and OL uncertainty trial types) on the model-generated data. First, examining the effect of past outcome (EL effect) and of past partner's action (OL effect) on choice (Fig. 4A, B), we found that as expected, the EL effect was well recovered by the EL model and both arbitration models, while the OL effect was well recovered by the OL model and both arbitration models. The baseline model was not able to recover any EL or OL learning effect. Correlations between the data and the model predictions across individuals confirmed this result (Fig. 4C, D), with the EL model accurately predicting the EL but not OL effect, the OL model accurately predicting the OL but not EL effect, and the dynamic arbitration model accurately predicting both effects. Second, we predicted that the uncertainty effects, i.e. the extent to which each strategy use varies with EL and OL trial uncertainty, should be appropriately recovered by the dynamic arbitration model, since this is the only model that explicitly modulate strategy weights based on uncertainty. And indeed, we found that the interactions between strategy use and uncertainty in data generated by the dynamic arbitration model (Fig. 4E–H right) matched those observed in the data (Fig. 4E–H left), with the model showing a clear effect of OL trial uncertainty on the OL effect (Fig. 4E, G) and of EL trial uncertainty on the EL effect (Fig. 4F, H). Correlations between the data and model predictions across individuals also showed strong recovery for the effect of uncertainty on the corresponding strategy (change in EL effect for low vs high EL uncertainty trials – data vs model predictions: Study 1: $R(126) = 0.795$, $P < 0.001$, Study 2: $R(493) = 0.870$, $P < 0.001$; change in OL effect for low vs high OL uncertainty trials – data vs model predictions: Study 1: $R(126) = 0.867$, $P < 0.001$, Study 2: $R(493) = 0.886$, $P < 0.001$; Fig. S5A–D). Interestingly, we also found that when running that same posterior predictive check analysis with the *condition* definition of OL and EL uncertainty (instead of the *trial* definition), the predictions of the dynamic arbitration model were not as strongly correlated with the data (change in EL effect for low vs high EL uncertainty condition – data vs model predictions: Study 1: $R(126) = 0.588$, $P < 0.001$, Study 2: $R(493) = 0.712$, $P < 0.001$; change in OL effect for low vs high OL uncertainty condition – data vs model predictions: Study 1: $R(126) = 0.633$, $P < 0.001$, Study 2: $R(493) = 0.593$, $P < 0.001$; Fig. S5E–H). This further validates the uncertainty trial definitions shown in Fig. 1B. Finally, we also found that dynamic arbitration weight values extracted for each participant from the dynamic arbitration model varied as predicted according to these trial definitions (Fig. S6).

## Group differences in learning, mixture of strategies and arbitration

To assess the behavioral relevance of this classification of participants in groups according to each individual best-fitting model and to further characterize the underlying heterogeneity, we calculated the

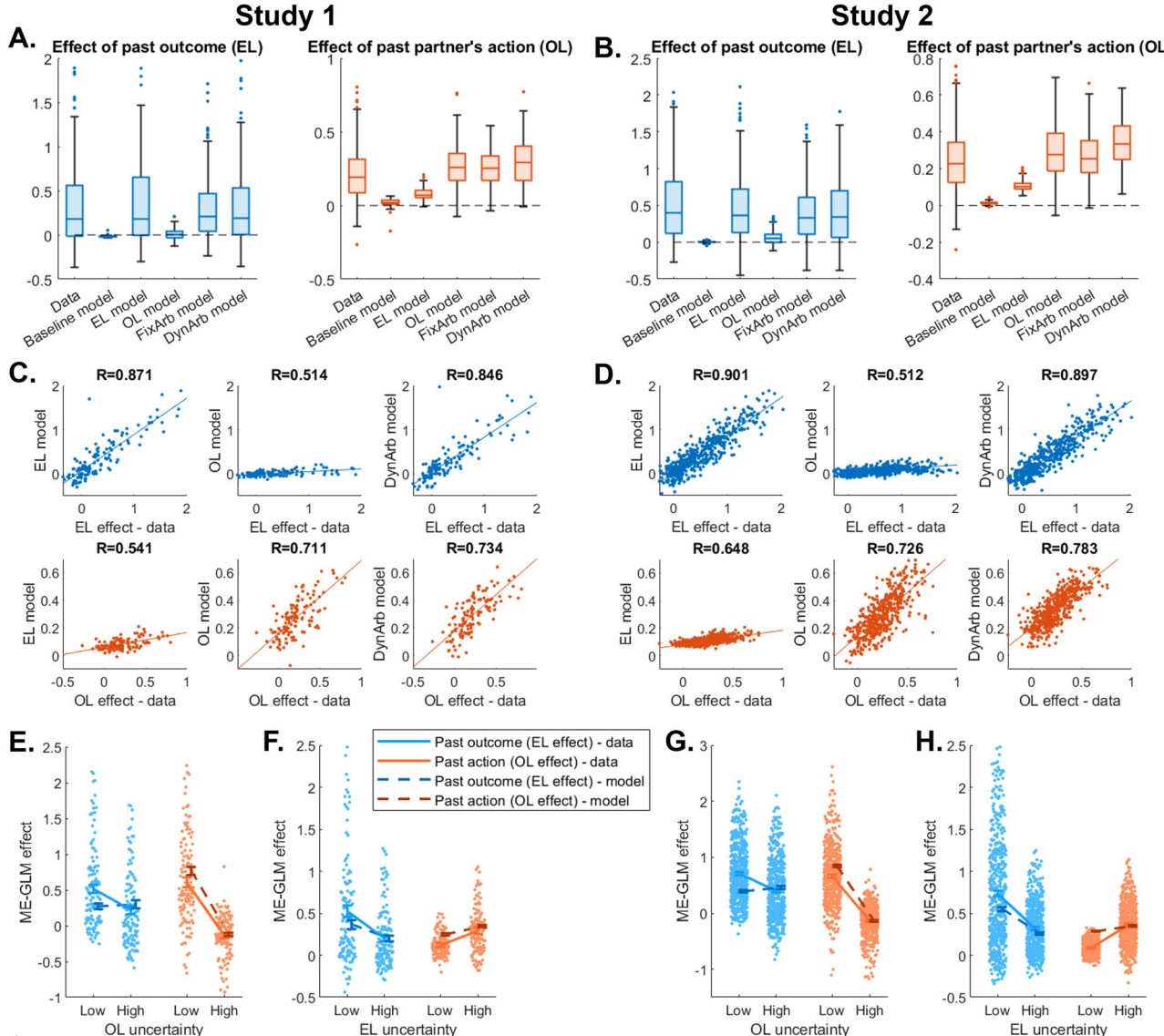

**Fig. 4 | Posterior predictive checks of strategy-specific effects.** The mixed-effects generalized linear model (ME-GLM) predicting choice from past outcome (experiential learning (EL) effect, blue) and past partner's action (observational learning (OL) effect, orange) was run on choice data generated with each of the 5 models, using participants' best-fit parameters. **A**, **B** Plotted are boxplots of the resulting individual random effects for Study 1 (**A**) and for Study 2 (**B**). Random effects obtained on the actual data are shown on the left-most box plot for comparison. Horizontal lines represent the median, boxes represent the inter-quartile range, whiskers range from the minimum to maximum value excluding outliers, which are shown as individual dots. **C**, **D** Individual random effects obtained from participants' data (x-axis) and from model-generated data (y-axis) are shown for

Study 1 (**C**) and for Study 2 (**D**), together with the best-fit linear regression line and correlation coefficient R, for the EL model (left), OL model (middle) and dynamic arbitration model (right). **E**–**H** To ensure the dynamic arbitration model can reproduce the effect of uncertainty on behavior, we ran the ME-GLM on participants' choices and on choices generated by the dynamic arbitration model separately for low and high OL uncertainty (Study 1: **E**, Study 2: **G**) and for low and high EL uncertainty (Study 1: **F**, Study 2: **H**). Data depicts the fixed effect coefficient estimates for low and high uncertainty (solid lines: data, dashed lines: model predictions); error bars are the standard error of the ME-GLM coefficients; dots are individual random effects from the data. Study 1: $N = 126$ independent participants (**A**, **C**, **E**, **F**); Study 2: $N = 493$ independent participants (**B**, **D**, **G**, **H**).

behavioral metrics shown in Figs. 2, 3, and compared them across groups, separately for each study. We also performed posterior predictive checks on the data split by actual groups, or by randomly assigning participants into groups, to ensure our behavioral differences between groups were appropriately recovered by the models. In all statistical analyses, we additionally controlled for individual differences in gender, age, education level and cognitive ability scores (from the ICAR) to ensure those factors could not explain any differences between groups (see Table S4 for all mixed-effect models equations and statistics). We note that there were some group differences in some of these variables (see Fig. S7 for details), hence the necessity to ensure our results were robust to controlling for them. Note also that

the sample size in those analyses was slightly reduced, given missing data ($N = 125$ in Study 1 because of one participant missing ICAR score; and $N = 489$ in Study 2 because of four participants missing education level).

First, we found that calculating the five models' out-of-sample accuracy separately for each group confirmed that each group was best fit by its respective model (Fig. 5A, B). Then, to compare the learning curves (Fig. 5C, D, Table S4A), we ran a linear mixed effect model predicting accuracy from the interaction between trial since last reversal (varying from 1 to 8) and group, controlling for the covariates of no interest described above and with a random intercept. We found a significant interaction between trial and group (Study 1:

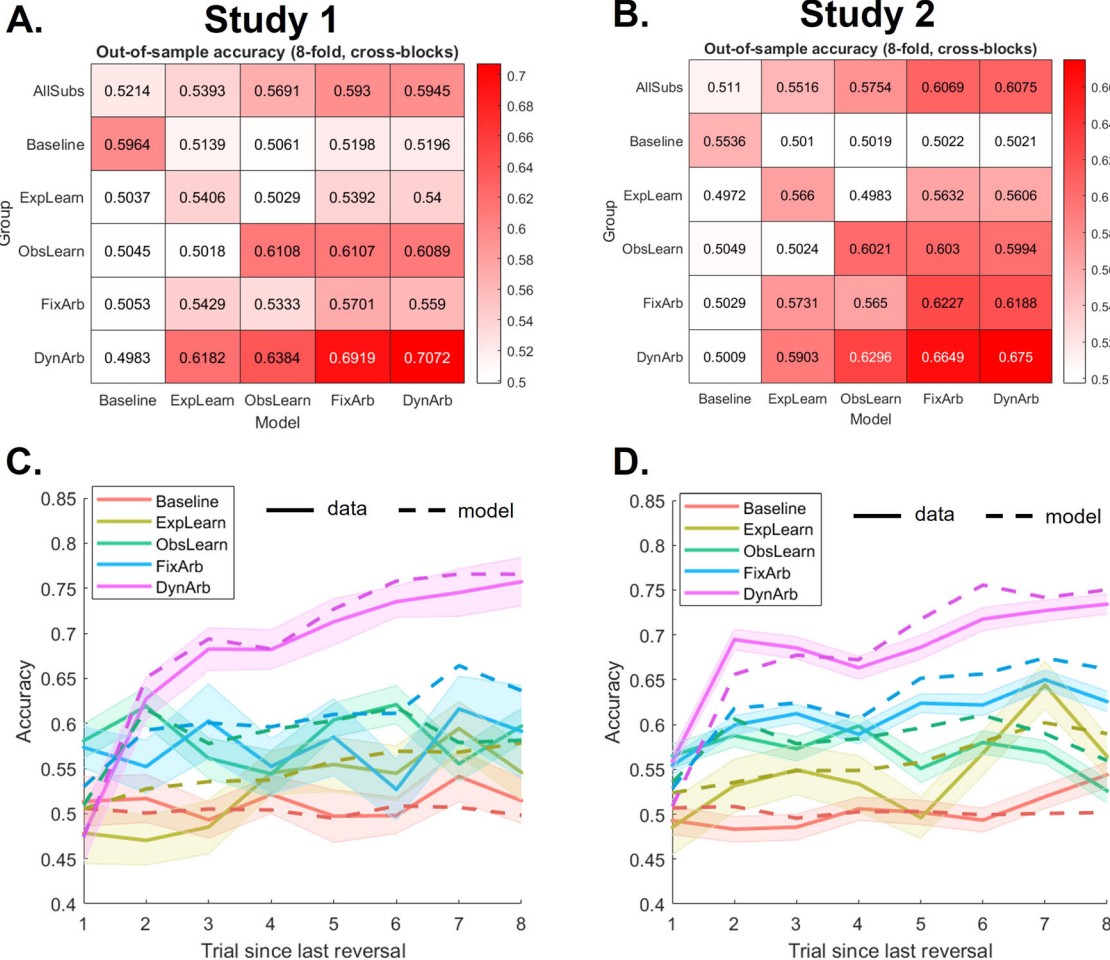

**Fig. 5 | Model out-of-sample predictive accuracy and learning curves by group.**
**A**, **B** We computed out-of-sample accuracy for each participant across blocks (leaving one block out) and for each of the five models, in Study 1 (**A**) and Study 2 (**B**). The top row of the heatmap shows the average predictive accuracy for each model, while the bottom five rows show the breakdown for each group. **C**, **D** Mean

learning curves (similar to Fig. 2A, B) were computed from participants' data separately for each group (solid lines, see Table S4A for statistics), and from model-generated data using each group's best-fitting model (dashed lines), in Study 1 (**C**) and Study 2 (**D**). The shaded area represents standard errors across participants within each group.

F(4,875) = 13.49, *P* < 0.001, $\eta_p^2$ = 0.058; Study 2: F(4,3423) = 18.81, *P* < 0.001, $\eta_p^2$ = 0.022), suggesting learning differences between groups, such that people in the baseline group essentially show no learning, while those in the dynamic arbitration group show the steepest learning curve (95% CI for the effect of trial in the dynamic arbitration group compared to the baseline group: Study 1: [0.021, 0.042]; Study 2: [0.006, 0.017]). Learning curves generated from the four learning models with completely hypothetical parameters (i.e. not the best-fitting parameters) confirmed that learning occurs in those models, and that the dynamic arbitration model produced the fastest learning (Fig. S8A). Examining learning curves generated from each group's best-fitting model (using individual participants' best-fitting parameters within that group) showed an almost perfect match to the data (solid vs dashed lines in Fig. 5C, D), which was not observed when group membership was randomly shuffled (Fig. S8B, C).

To compare the GLM betas measuring EL and OL contributions to behavior (Fig. 6, Table S4B), we ran a linear mixed effect model predicting the mixed effect GLM coefficient values from the interaction between effect type (past outcome versus past partner action) and group, also controlling for covariates and with a random intercept. We found a significant interaction between effect type and group (Study 1: F(4,125) = 25.55, *P* < 0.001, $\eta_p^2$ = 0.450; Study 2: F(4489) = 73.73, *P* < 0.001, $\eta_p^2$ = 0.376). The interaction was mostly explained by different drivers of behavior in the ExpLearn and ObsLearn groups.

Specifically, as expected, people in the ExpLearn group relied more strongly on past outcome (EL effect) than past partner action (OL effect) to guide behavior (paired two-tailed t-test – Study 1: t(20) = 3.16, *P* = 0.005, *d* = 0.706, 95% CI [0.084, 0.410]; Study 2: t(23) = 4.27, *P* < 0.001, *d* = 0.89, 95% CI [0.179, 0.514]), while behavior in the ObsLearn group was more strongly driven by past partner action than past outcome (Study 1: t(39) = 3.60, *P* < 0.001, *d* = 0.577, 95% CI = [0.052, 0.184]; Study 2: t(94) = 3.07, *P* = 0.003, *d* = 0.316, 95% CI [0.020, 0.092]). We also found a main effect of group (Study 1: F(4,125) = 40.74, *P* < 0.001, $\eta_p^2$ = 0.566; Study 2: F(4,489) = 106.4, *P* < 0.001, $\eta_p^2$ = 0.769), driven as expected by overall weakest EL and OL effects in the baseline group, consistent with no learning, but also by overall strongest EL and OL effects in the dynamic arbitration group (95% CI of DynArb vs Baseline group difference: Study 1: [0.889, 1.196], Study 2: [0.750, 0.900]). The latter can be explained by higher overall accuracy in the dynamic arbitration group, combined with positive correlations between accuracy and strength of both EL and OL effects (Accuracy & EL effect: Study 1: R(126) = 0.662, *P* < 0.001, Study 2: R(493) = 0.723, *P* < 0.001; Accuracy & OL effect: Study 1: R(126) = 0.895, *P* < 0.001, Study 2: R(493) = 0.896, *P* < 0.001). Additionally, posterior predictive checks confirmed that pattern of GLM effects between groups, whereby GLM effects from model-generated data matched the data well when split by actual groups (darker colored bars in Fig. 6), but not when split by randomly shuffled groups (grey bars in Fig. 6).

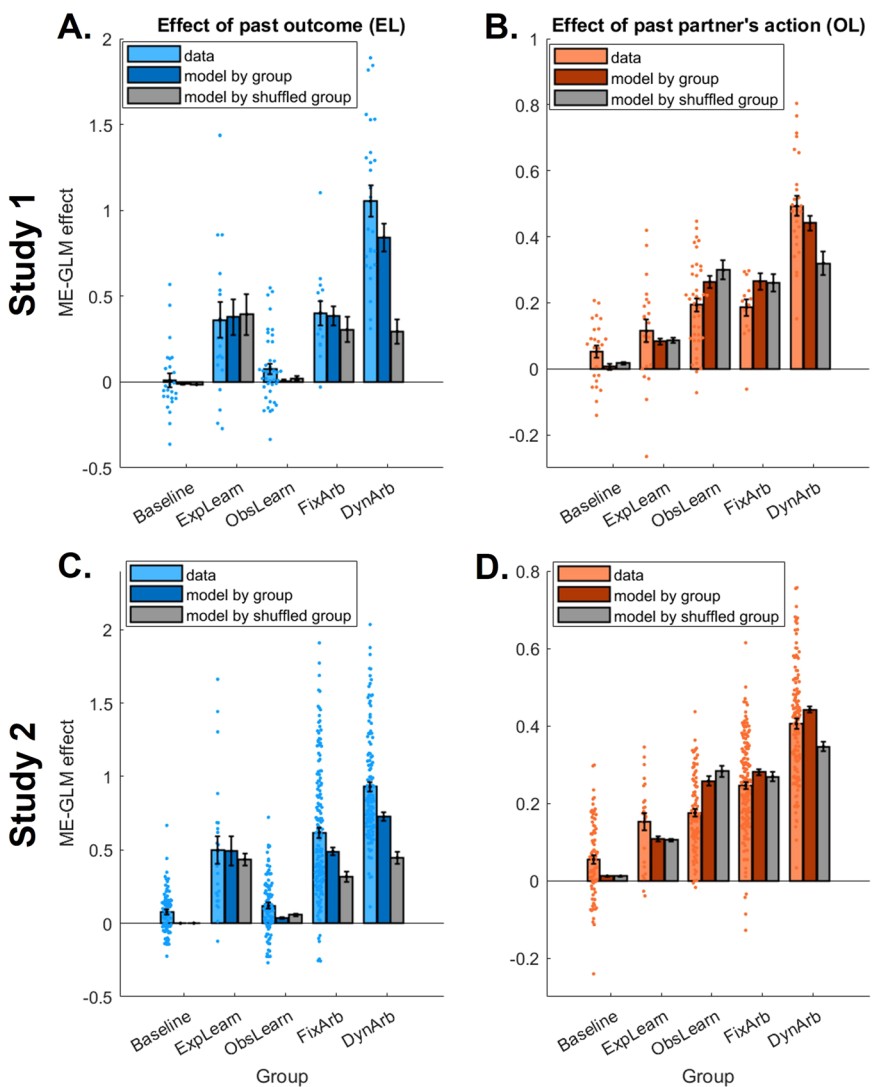

**Fig. 6 | Group differences in single strategy use and associated model predictions.** The main effects of past outcome (EL effect, **A, C**) and the effect of past partner's action (OL effect, **B, D**) on choice (previously shown in Fig. 2E-F) are now calculated separately for each group for Study 1 (**A-B**, $N_{Baseline} = 25$, $N_{ExpLearn} = 21$, $N_{ObsLearn} = 40$, $N_{FixArb} = 14$, $N_{DynArb} = 26$) and Study 2 (**C, D**, $N_{Baseline} = 83$, $N_{ExpLearn} = 24$, $N_{ObsLearn} = 95$, $N_{FixArb} = 160$, $N_{DynArb} = 131$). Light colored bars represent the mean mixed-effects generalized linear model (ME-GLM) coefficients from participants' data (see Table S4B for statistics), whereby each dot is an individual participant (random effect). Dark colored bars represent the ME-GLM coefficients from data generated by each of the 5 models using participants' best-fitting parameters, then showing the mean effect for each group using that group's best-fitting model. Grey bars similarly represent the mean ME-GLM coefficients from model-generated data, but after assigning participants into the 5 groups at random then using each group's corresponding model. Error bars represent standard errors of the ME-GLM coefficients.

Finally, we compared the behavioral signatures of arbitration by highlighting group differences that dissociate participants expressing a fixed mixture versus a dynamic arbitration between strategies. Based on the breakdown of OL choice propensity per trial type (shown on Fig. S2A, B), we computed a behavioral index of arbitration as the difference in OL choice propensity between trials where OL is expected to be most preferred (trials with low OL uncertainty, high EL uncertainty, low reward magnitude) and trials where EL is expected to be most preferred (trials with high OL uncertainty, low EL uncertainty, high reward magnitude). This difference is depicted with the green arrow on Fig. S2A, B. We then calculated this index separately for each group and found a significant effect of group on arbitration index in a linear model controlling for all covariates (Study 1: F(4,112) = 17.13, P < 0.001, $\eta_p^2 = 0.380$, Fig. 7A; Study 2: F(4,465) = 80.55, P < 0.001, $\eta_p^2 = 0.409$, Fig. 7B). Specifically, arbitration was found to be maximal in the dynamic arbitration group and significantly larger than in the fixed mixture group (Welch two-sample t-test assuming unequal variance;

Study 1: t(17.23) = 4.74, P < 0.001, d = 1.70, 95% CI [0.252, 0.656]; Study 2: t(277.44) = 7.15, P < 0.001, d = 0.836, 95% CI [0.172, 0.303]), suggesting a behavioral dissociation between the two arbitration groups, whereby dynamic arbitration is associated with a more extreme variation in strategies according to the conditions of the environment.

To further examine the effect of each uncertainty trial type on each strategy separately, we also analyzed how the random effects of past outcome (EL) and past partner action (OL), estimated separately for each uncertainty trial type (and shown on Fig. 3C, D), differed between groups. For each trial type (OL uncertainty, EL uncertainty), we ran a linear mixed model predicting the random effect from an interaction between uncertainty trial type (high, low), strategy (EL, OL) and group, controlling for covariates and including a random intercept (Table S4C, D). We found a significant 3-way interaction for each manipulation, and for both Study 1 (Fig. 7C: OL uncertainty * strategy * group, F(4,375) = 20.22, P < 0.001, $\eta_p^2 = 0.177$; Fig. 7D: EL uncertainty * strategy * group, F(4,375) = 19.90, P < 0.001, $\eta_p^2 = 0.175$) and Study 2

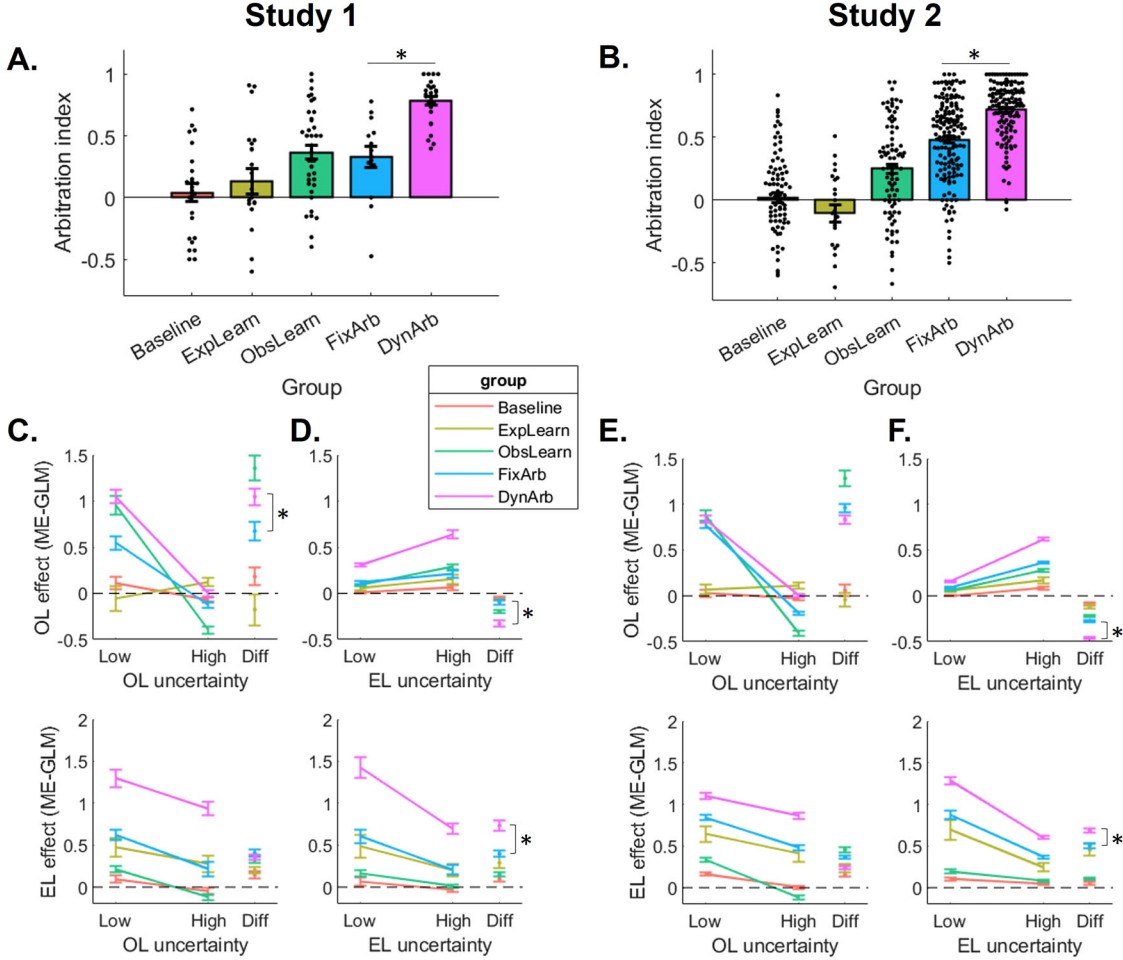

**Fig. 7 | More extreme signatures of arbitration in the dynamic arbitration group. A, B** An index of arbitration was calculated as the difference in the propensity to choose according to observational (OL) vs experiential (EL) learning between trials where OL should be most favored and trials where EL should be most favored – see green arrows on Fig. S2A, B for an illustration. This arbitration index was then calculated and averaged separately for each group, in Study 1 (**A**, $N_{Baseline}$ = 25, $N_{ExpLearn}$ = 21, $N_{ObsLearn}$ = 40, $N_{FixArb}$ = 14, $N_{DynArb}$ = 26) and Study 2 (**B**, $N_{Baseline}$ = 83, $N_{ExpLearn}$ = 24, $N_{ObsLearn}$ = 95, $N_{FixArb}$ = 160, $N_{DynArb}$ = 131). Significance was assessed through a linear regression predicting arbitration index from group and controlling for covariates, followed by a two-sided t-test to specifically compare the dynamic arbitration and fixed mixture groups (**A:** t(17.23) = 4.74, 95% CI [0.252, 0.656], P < 0.001; **B:** t(277.44) = 7.15, 95% CI [0.172, 0.303], P < 0.001). Each dot is an individual participant. **C–F** The random effects obtained from the analyses presented in Fig. 3C, D were averaged separately for each group, for Study 1 (**C, D**) and Study 2 (**E, F**), thus showing the effect of OL

uncertainty trial type (**C, E**) and EL uncertainty trial type (**D, F**) on the GLM effects of past partner action (OL effect; top) and past outcome (EL effect; bottom). See Table S4C, D for GLM statistics. For each manipulation, the difference between high and low uncertainty trials is also depicted, allowing for a direct comparison of each arbitration signature between groups. Error bars represent SEM for each group. Two-sided t-tests were run to specifically test whether the dynamic arbitration group (magenta lines) exhibited a more extreme signature of arbitration than the fixed arbitration group (blue lines), with significant differences observed in the effect of OL uncertainty on OL in Study 1 (t(32.38) = 2.81, 95% CI [0.103, 0.644], P = 0.008, **C**), the effect of EL uncertainty on both strategies in Study 1 (OL: t(37.02) = −5.39, 95% CI [−0.323, −0.147], P < 0.001, **D** top; EL:t(36.29) = 4.86, 95% CI [0.196, 0.476], P < 0.001, **D** bottom), and in Study 2 (OL: t(261.11) = −11.08, 95% CI [−0.458 −0.273], P < 0.001, **F** top; EL: t(288.18) = 4.67, 95% CI [0.103, 0.253], P < 0.001, **F** bottom).

(Fig. 7E: OL uncertainty * strategy * group, F(4,1467) = 33.50, P < 0.001, $\eta_p^2$ = 0.084; Fig. 7F: EL uncertainty * strategy * group, F(4,1467) = 97.38, P < 0.001, $\eta_p^2$ = 0.210). This showed that the influence of uncertainty trials on the signature of EL vs OL further varied across groups. More specifically, OL uncertainty trials primarily influenced the effect of past partner action (OL signature) and did so more strongly for individuals in the ObsLearn and arbitration groups. In contrast, EL uncertainty trials primarily influenced the effect of past outcome (EL signature) and did so more strongly for individuals in the dynamic arbitration group, compared to all other groups. This last result is particularly noteworthy as it suggests that it is mostly EL arbitration (i.e. arbitration driven by EL uncertainty) that differentiates between the fixed and dynamic arbitration groups, rather than OL arbitration (i.e. arbitration driven by OL uncertainty), which seems to be exhibited in both arbitration groups.

## Relevance of groups for psychopathology

Having established that the groups defined based on model-fitting displayed the expected differences in behavioral signatures of interest (learning, reliance on OL vs EL, and arbitration), we set out to explore whether the five groups also differed in meaningful ways on a range of transdiagnostic symptom dimensions. Given our hypothesized link between strategy used and symptom dimensions relevant to anxiety, social anxiety, and autism, we collected four questionnaires (State-Trait Anxiety Inventory, Beck Depression Inventory, Liebowitz Social Anxiety Scale, and Social Responsiveness Scale, see Methods for details). To extract underlying symptom dimensions and reduce collinearity between summary scores on those scales, we first ran a factor analysis on the individual item scores from the questionnaires, pooled across the two studies to ensure sufficient power to run the factor analysis (N = 568). We first determined the optimal number of factors

by running the factor analysis from 1 to 20 factors and selected the number of factors among this set that provided the lowest BIC (Fig. S9A). We found that 8 factors provided the best-fitting solution (BIC = −59393, BIC difference from other number of factors > 14, Tucker-Lewis index = 0.745, Root Mean Square Error of Approximation = 0.042, fit = 0.967). The factor loadings suggest the following transdiagnostic symptom interpretation associated with each factor (Fig. S9B). Factor 1 reflected depressive symptoms, with highest loadings on all BDI items, as well as some of the STAI-Trait items such as failure, unhappiness, and dissatisfaction. Factor 2 reflected heightened social anxiety, loading on a majority of items from the LSAS. Factor 3 reflected autism-like traits, loading on a majority of items from the SRS. Factor 4 reflected state anxiety symptoms, loading mostly on STAI-State items. Factor 5 reflected poor social responsiveness, loading specifically on positively scored SRS items. Factor 6 loaded on items from both SRS and LSAS that reflect social group avoidance. Factor 7 reflected trait anxiety, loading mostly on STAI-Trait items that relate to disturbing or obsessive thoughts. Finally, Factor 8 reflected traits associated with performance anxiety, loading most strongly on LSAS items such as acting, speaking up, reporting to a group, etc. Factors were allowed to correlate given the *oblimin* rotation, but correlations between factors remained low enough to ensure unique variance attributed to each factor, ranging from R(568) = 0.08 to R(568) = 0.56 (Fig. S9C). These correlations were overall much lower than the correlations between the questionnaire scores, ranging from R(568) = 0.45 (between STAI-State and LSAS) to R(568) = 0.82 (between STAI-Trait and BDI), thus justifying the factor analysis approach to better identify separate symptom dimensions.

To assess whether the five groups, defined based on their learning strategy on the task, differed on those 8 symptom dimensions, we ran a linear mixed-effects model predicting the factor scores (each factor representing a symptom dimension) from an interaction between symptom dimension and group, including a random intercept, and controlling for gender, age, education, ICAR score, as well as study group (given that we pooled data from both studies). We found a significant interaction (F(28,3948) = 2.38, $P < 0.001$, $\eta_p^2 = 0.017$, Fig. 8A–E, Table S5A), suggesting that the groups differed in their symptom dimensions. For comparison purposes, results from the same analyses with the 5 questionnaire summary scores, instead of the 8 separable symptom dimensions, are shown in Table S5B.

Post-hoc tests using R's *emmeans* function to compute marginal means highlighted the following drivers of the interaction. First, we examined differences in symptom dimensions within each group, using Tukey method p-value adjustment for comparing a family of 8 estimates (the number of symptom dimensions), revealing significant effects in two of the groups. In the baseline group (Fig. 8A), individuals were characterized by high autistic traits, poor social responsiveness, and low trait anxiety (autistic traits vs trait anxiety: estimate=0.527, t(3983) = 4.47, $P = 0.002$, $d = 0.648$, 95% CI [0.364, 0.933]; social responsiveness vs trait anxiety: estimate=0.411, t(3983) = 3.49, $P = 0.012$, $d = 0.506$, 95% CI [0.221, 0.790]). In the dynamic arbitration group (Fig. 8E), individuals were characterized with the opposite pattern, that is low autistic traits, good social responsiveness, but high trait anxiety (autistic traits vs trait anxiety: estimate = −0.340, t(3983) = −3.51, $P = 0.011$, $d = −0.418$, 95% CI [−0.652, −0.185]; social responsiveness vs trait anxiety: estimate = −0.300, t(3983) = −3.10, $P = 0.041$, $d = −0.369$, 95% CI [−0.603, −0.135]). Second, we ran the complementary analysis, examining differences between groups for each symptom dimension, using Tukey adjustment for the 5 groups. We found differences in autistic traits between the baseline and fixed arbitration groups (t(3070) = 3.37, $P = 0.007$, $d = 0.509$, 95% CI [0.213, 0.805], Fig. 8A vs D), and between the observational learning and the fixed arbitration group (t(3095) = −2.73, $P = 0.049$, $d = 0.381$, 95% CI [0.108, 0.655], Fig. 8C vs D). Groups also differed on trait anxiety,

specifically between the baseline and dynamic arbitration groups (t(2847) = 4.05, $P < 0.001$, $d = 0.642$, 95% CI [0.331, 0.953], Fig. 8A vs E).

Overall, this suggests that the five model-based groups can be differentially characterized along two symptom dimensions: autistic traits and trait anxiety (Fig. 8F). We note that the two symptom dimensions were positively correlated across participants (R(568) = 0.16, $P < 0.001$, Fig. 8G), such that on average across the entire sample, individuals with high autistic traits also tend to score high on trait anxiety. Yet, we find that groups, especially the baseline and dynamic arbitration groups, differ significantly on these dimensions, suggesting that our model-based classification can help separate symptom dimensions that tend to coexist in the population.

## Discussion

Our aim in this study was two-fold: first, to test a computational account of reliability-driven arbitration between two domains, namely experiential learning (EL) and observational learning (OL); and second, to characterize the heterogeneity in strategy use, both in key signatures of behavior and in transdiagnostic symptom dimensions relevant to affective and social function.

To address the first aim, we designed a task in which the reliability of EL and OL were manipulated by means of changes in uncertainty conditions, resulting in key trials that could be clearly classified as high EL reliability trials, low OL reliability trials, and vice versa. Behavioral findings indicated that people clearly modulated their behavior in an expected way according to the reliability of each strategy, favoring EL when EL reliability was high and OL reliability was low, and favoring OL when OL reliability was high and EL reliability was low. Computational modelling confirmed this finding, showing that those participants who were best fit by our proposed dynamic arbitration model also exhibited the greatest reliability-driven modulation of behavior. Reliability in our model was defined using absolute prediction errors associated with each strategy as an index of uncertainty (or unreliability). This arbitration signal is consistent with the algorithmic and neural implementation of mixture of experts models in the literature[20], though future work is needed to further explore whether other implementations of reliability could perform better. In particular, this could be achieved through more optimized task designs that fully allow distinguishing between different reliability computations, which was outside the scope of the current study. Our analyses do however provide insights into how this dynamic arbitration mechanism differs from a fixed mixture model, which was originally proposed in early investigations of model-based/model-free arbitration during EL[52,53]. Not only did model recovery analyses show that the two arbitration schemes can be clearly differentiated, but behavioral signatures associated with each model pointed towards more 'extreme' signature of uncertainty-driven arbitration between EL and OL. In sum, a learner using a fixed mixture model will still be sensitive to trial-by-trial changes in uncertainty, since those variations will be captured in the value difference, and hence in the choice probabilities; however, using the proposed dynamic arbitration mechanism helps push this sensitivity to the extreme, leading to improved performance. Consistent with cross-domain arbitration, and with previous literature showing that humans do integrate social and experiential information when learning and making decisions[15,26–32,54], our findings also suggest that the fixed and dynamic arbitration groups (a substantial proportion of our sample) performed this task by integrating the predictions of both EL and OL. Only one of these studies in particular demonstrated the possibility of a dynamic, volatility-driven, arbitration between individual and social learning[54]. Although the individual learning used in that study was similar to our EL model in the current study (outcome of a binary lottery), the social learning component was quite different (learning from advice, rather than learning from observing another person's choices). Our findings thus further extend the concept of arbitration, via a reliability-weighted mixture of experts[20], to apply

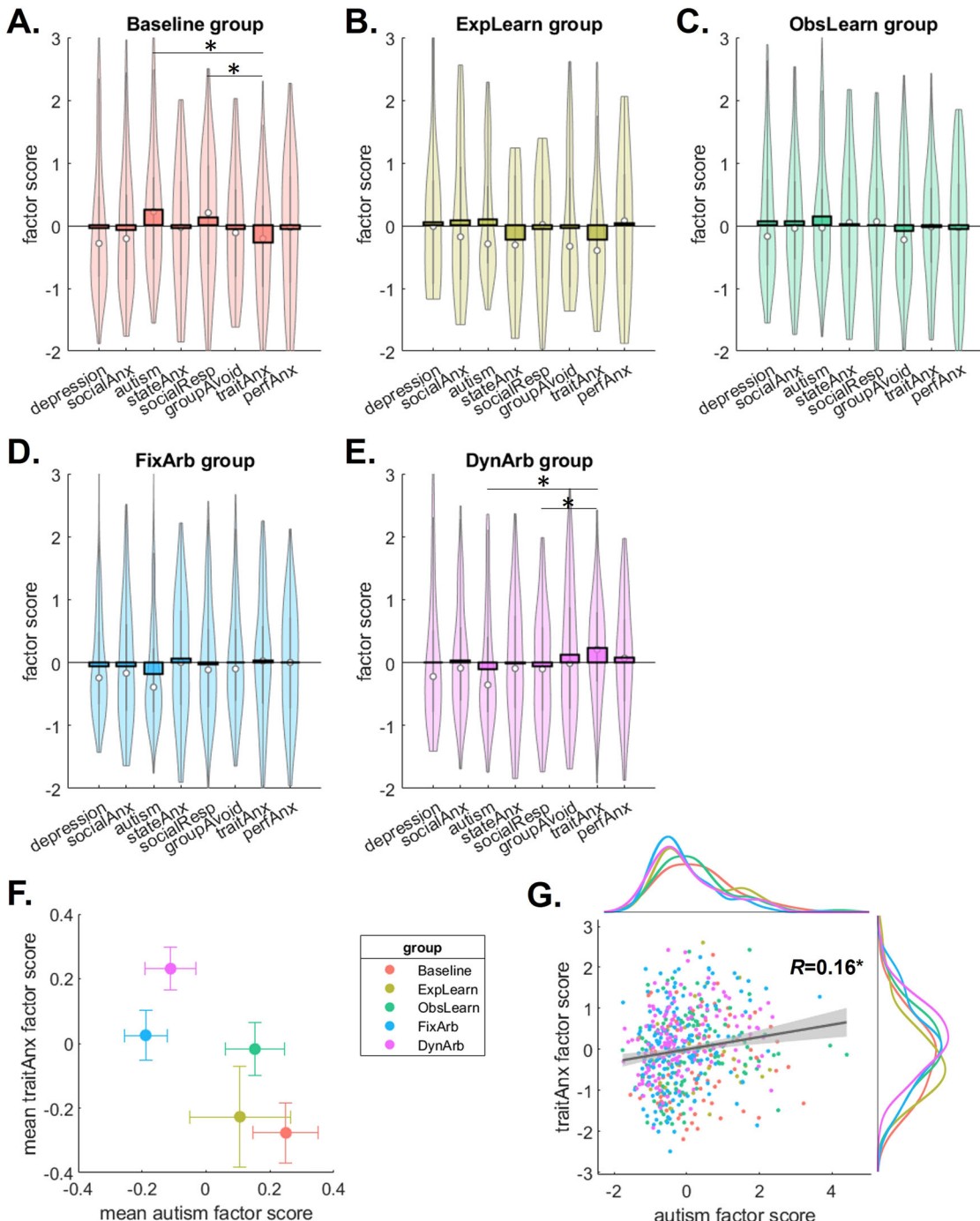

**Fig. 8 | Group differences in transdiagnostic symptom dimensions. A–E** A factor analysis on the questionnaire items yielded 8 factors representing separate symptom dimensions. Plotted are violin plots of the factor scores for each of the 5 groups (bars represent the mean, white dot represent the median, thicker grey error bars represent the interquartile range): (**A**) baseline group (*N* = 96), (**B**) experiential learning group (*N* = 38), (**C**) observational learning group (*N* = 125), (**D**) fixed arbitration group (*N* = 165), (**E**) dynamic arbitration group (*N* = 144). A linear mixed-effects model was run to test the significance of a factor*group interaction (see Table S5A for statistics), suggesting that factor scores differ significantly across groups, while controlling for gender, age, education level, cognitive ability and study group. Pairwise two-sided t-tests were run on the resulting marginal means to characterize the differences between factors within each group, using the Tukey method to correct for multiple comparisons. Significant differences were observed in the Baseline group (autism vs trait anxiety: t(3983) = 4.472, *P* < 0.001; social responsiveness vs trait anxiety: t(3983) = 3.487, *P* = 0.012; **A**) and in the DynArb group (autism vs trait anxiety: t(3983) = −3.508, *P* = 0.011; social responsiveness vs trait anxiety: t(3983) = −3.095, *P* = 0.042; **E**). **F, G** Follow-up analyses focusing on the two factors that account for the largest group differences: autism (factor 3) and trait anxiety (factor 7). **F** Mean and standard errors for the two factors by group indicate a dissociation between the two symptom dimensions. **G** Scatter plot with histograms of the two factor scores, colored by group, with a regression line across all participants (*N* = 568, two-sided Pearson's correlation: R = 0.16, *P* < 0.001).

across observational and experiential domains, rather than within-domain only (e.g. ref. 21 for EL; ref. 18 for OL). This dichotomy goes beyond the well-known model-based/model-free arbitration[21,22,55] given that the OL strategy is implemented in the absence of outcomes, that is by simply inferring the current goal from observing the other agent's actions. In that sense, outcomes cannot be used to reinforce cached values (like in model-free learning or in the EL strategy here) or to build a model of the world (model-based learning).

Overall, our model comparison also suggests that OL as its own strategy was favored relative to EL. It is possible that the OL and EL strategies aren't matched in terms of complexity, working memory demands, or cognitive processes at play, and that OL may require additional demands on the learner's part. That said, our finding that across both studies participants tend more towards using OL compared to EL mitigate this concern about complexity differences. Additionally, our focus on an arbitration mechanism does mean that the strategies being arbitrated have to be dissociable and rely on separable computations, in order to best characterize how arbitration is governed. Our task achieved this dissociation, with OL and EL updates taking place at different timepoints in the trial, OL and EL predictions being only weakly correlated, and OL and EL uncertainty trials being separable. As part of our study design, we decided to focus on uncertainty as the main factor driving the reliability of each strategy. We manipulated uncertainty across different blocks, with EL uncertainty levels varying within each OL uncertainty block to allow for the possibility of slower learning during OL. The data did not confirm this (if anything learning was faster during OL than EL), suggesting that future designs could better ensure uncertainty periods are symmetrical across strategies. That said, the uncertainty conditions were not used in the analyses since we identified that our specific trial-by-trial definitions of uncertainty were better predictors of choice than the uncertainty conditions, thus mitigating the design asymmetry concern. We note, however, that our trial definition of uncertainty was based on the immediately preceding sequence of trials and could have involved a more sophisticated definition of uncertainty over longer timescales. In addition, it is possible that other factors than uncertainty could drive arbitration. For example, OL reliability may be influenced by observing different partners who vary in performance and/or expertise, or by manipulating the social context (cooperative versus competitive interaction, incentive for the partner to deceive the participant or to act with a different goal in mind). EL reliability could also vary as a function of stakes being manipulated in a more meaningful way than in the current design, or by increasing or decreasing the ability to experientially sample the tokens and their outcomes. Another interesting question would also be to assess how the reliance on each strategy, and arbitration between them, may vary in response to more naturalistic and ecologically valid ways of implementing the strategies in the task. Future work is needed to better characterize the role these factors may play in cross-domain arbitration.

To address our second aim of characterizing heterogeneity, we leveraged the possibility of collecting large-scale datasets online. Our modelling results revealed that no 'winning' model explained all (or a majority of) participants' data best; rather, different groups of participants were found to rely on different strategies to solve this task. Specifically, while a proportion of participants relied on our proposed dynamic arbitration model, or on a fixed mixture of EL and OL, as described above, some participants were also found to use a single strategy (EL-only or OL-only), while a small proportion of participants were best characterized by a baseline model incorporating irrelevant non-learning strategies. Participants in each group were clearly characterized by unique behavioral signatures. For example, participants who relied primarily on EL (ExpLearn group) exhibited a stronger effect of past outcomes on their current choice (signature of EL) relative to the ObsLearn group, while the opposite group difference was found when examining the effect of past partner's action on

current choice (signature of OL). The extent to which those effects differed between trials with high versus low EL and OL uncertainty was also found to be more extreme in the Dynamic arbitration group, compared to the fixed arbitration group. The dynamic arbitration model was built to be the most advantageous strategy to solve this task. We confirmed that this was the case using simulated data and posterior predictive checks, showing stronger signatures of learning and of both EL and OL strategy used in data generated by the dynamic arbitration model compared to other models. This explained why participants best fit by the dynamic arbitration models were overall better learners, and reproduced the heterogeneity in behavioral patterns observed across groups. Finally, none of these behavioral signatures were present in the Baseline, non-learner group. While the baseline model is not a relevant strategy to perform this task, including this model in our set allowed us to characterize behavior above chance for a substantial proportion of the participants (about 20% in Study 1 and 15% in Study 2) for whom actual learning models would have performed at chance. Therefore, this enabled us to keep these participants included in the analyses rather than excluding them for poor performance, a common technique used to diminish the noise and improve data quality in online studies[35,56]. We hope that this method of characterizing non-learning behavior, rather than simply excluding participants, will become more widespread in analyses of online datasets going forward, especially when considering the relevance for psychopathology, whereby 'poor' performance may be indicative of symptoms of interest. Indeed, such task-based widespread exclusions could lead to a reduction in the range of relevant symptom dimensions that may be associated with the use of non-learner strategies. That said, we do acknowledge that the interpretability of behavior in this group is limited, given that more than one strategy was included in the model and that those strategies aren't necessarily reflective of underlying cognitive mechanisms. Contrary to the other groups for which associations with symptom dimensions can be directly interpreted in light of the theory-driven modelling approach, in this non-learning group it remains more challenging to characterize what associations might mean, whether they reflect a general deficit in learning, motivational impairments, or reduced working memory capacity, among others. Additionally, we note that depending on the sample, it may still be worth excluding participants who are not doing the task at all (e.g., high number of missed responses), repeatedly fail to pass a quiz the instructions (i.e. possibly indicative of being a bot), or fail attention checks during the questionnaires[57].

Finally, in further characterizing the heterogeneity in strategy use, we provide the first evidence that using such a strategy-based classification of participants carries relevance for psychopathology. We reduced the dimensionality of the questionnaire items into 8 main factors reflecting largely separate symptom dimensions relevant to social dysfunction, autistic traits, anxiety (trait, state, social, performance, avoidance), and depression. We find a significant interaction between factor and group in predicting symptom severity (indexed by higher values of factor scores), suggesting differences in symptom dimensions between the groups. Specifically, consistent with some of our predictions, we found that trait anxiety and autistic traits were the two factors accounting for most of the group differences. The strongest differences were between the Baseline group, characterized by high autistic traits and low trait anxiety, and the Dynamic arbitration group, exhibiting the opposite profile (low autistic traits and high trait anxiety). Importantly, these differences remained significant when controlling for additional covariates such as age, gender, education level and cognitive ability. Known difficulties in behavioral adaptation and flexibility during social inference in autism[47,58,59] are consistent with the observed group differences, such that those participants relying on more optimal or advantageous strategies (arbitration) also exhibit the lowest autism factor score, and those scoring high on autistic traits are more likely to use an irrelevant, non-learner strategy

during the task. Trait anxiety, on the other hand, has also been associated with difficulties in adapting learning to changes in uncertainty[38–40], which would also lead to the prediction of reduced reliance on dynamic arbitration in highly anxious individuals. Interestingly, we see the opposite pattern (highest trait anxiety factor score in the dynamic arbitration group). A few differences could account for this. First, it is possible that the effect shown in the literature is specific to volatility (i.e. frequency of the reward/transition probability reversals), which was primarily manipulated in the studies cited above, rather than changes in reward/transition probability itself, which was manipulated in the present task. Second, this effect may be emerging given that we are not only looking at trait anxiety summary score, but at a factor representing a symptom dimension with only a subset of questionnaire items. And indeed, when examining group differences in mean summary score from the STAI-Trait questionnaire, there were no statistically significant difference ($F_{(4, 563)} = 0.59$, $P = 0.671$, $\eta_p^2 = 0.004$), suggesting that this effect may only emerge when other correlated symptom dimensions (such as traits and symptoms associated with depression) are controlled for. This finding of heightened arbitration in anxiety could also be interpreted in light of a recent study demonstrating that high anxiety is associated with increased information-seeking in response to large changes in the environment[60], consistent with an ability to adapt behavior to changing contingencies, which is also needed during arbitration.

We also found no clear separability between the ObsLearn and ExpLearn groups using the factor scores. Although the largest differences between those two groups appear to be driven by the state and trait anxiety factors, those were not significant. This could be due to the relatively low sample size of the ExpLearn group even after pooling the two datasets (N = 38). Overall, having to pool the two datasets for running the factor analysis and testing group differences in factor scores is a possible limitation of the current study, which warrants replication in future large-scale studies. However, it is worth noting that this was necessary to ensure sufficient power for the factor analysis (as running the factor analysis on Study 1 data only would not have been possible), that this analysis of relevance for psychopathology was exploratory in nature, and that the study group was controlled for in these analyses. We also note that as a clear strength of the study, all other analyses characterizing heterogeneity in behavior and model fits were replicated in two completely independent samples.

Taken together, our findings demonstrate the relevance of our model-based grouping approach, whereby individual differences in best-fitting model are leveraged, rather than assuming a 'winning-model-fits-all' approach. This is consistent with previous work having shown different learning strategies across individuals, for example across age groups during development[61,62], or between clinical and non-clinical groups[63]. In the present study, we further emphasize the potential of this approach for separating symptom dimensions (here, autism and trait anxiety) that tend to coexist in the general population (see the positive correlation in the current sample) and in the clinic, whereby 40-70% of individuals with autism also meet criteria for at least one anxiety disorder[64,65]. Large datasets combined with computational methods, such as the approach proposed in the current study, are gaining importance to help characterize individual differences[66] and pave the way towards individualized diagnoses and treatment plans.

## Methods

### Participants

Participants were recruited online and anonymously through Prolific Academic (https://www.prolific.co/) across two independent studies. Study 1 (exploration sample) included 128 participants (56 females, 71 males, 1 non-binary; mean age = 32.84 years old ± 10.90 SD). Study 2 included 493 participants (290 females, 199 males, 4 non-binary; mean age = 28.48 years old ± 9.90 SD). Two participants were excluded from

Study 1 for missing more than 25% of trials. No participant in Study 2 met this exclusion criterion. Screening criteria applied included US as country of residence, fluency in English, age between 18 and 65, no literacy difficulty, normal or corrected-to-normal vision. Those criteria were assessed through Prolific's screening tools based on participants self-report. All participants provided informed consent. After an initial IRB review, the protocol used in both Study 1 and Study 2 was deemed exempt from full IRB review by the Caltech Institutional Review Board, due to being judged to be of minimal risk to participants and meeting several other criteria required for an exempt status. In Study 1, participants were paid $6.00 (US Prolific account) for their participation for 45 minutes, as well as a performance- and attention-based bonus of up to $2.00 (mean bonus = $1.09 ± 0.43 SD). In Study 2, participants were paid in British Pounds (UK Prolific account), specifically £5 (around $7) for 50 min and up to £1.50 (around $2) bonus (mean bonus = £0.78 ± 0.40 SD).

### Task

The task consisted of 8 blocks of 20 trials each. On each trial, participants first observed a partner choose between two boxes (represented by a fractal image) for 2.5 s, then were shown which of two tokens (blue or orange) the chosen box delivered for 1 s. It was then the participant's turn to choose directly one of the two tokens (in maximum 3 s) and receive the associated outcome for 1 s (Fig. 1A). Box and token left/right positions were randomized. Throughout the task, two probabilities dynamically varied (Fig. 1B, orange and blue lines): the probability that each box yields an orange vs blue token, and the reward probability associated with each token. The two boxes yielded opposite probabilities of the two token types. Specifically, in the low uncertainty condition the probabilities were 80/20 (one box yields 80% of orange tokens and 20% of blue token and the other box 80% of blue tokens and 20% of orange tokens). In the high uncertainty condition, the probabilities were 60/40. Those probabilities were assigned at the beginning of a block and kept constant throughout a block. On each block a new pair of boxes (new fractals) were used. Participants were not explicitly instructed about those contingencies; however, they were told that "one box has a higher proportion of blue tokens; the other one a higher proportion of orange tokens". The reward probability associated with each token also varied between periods of low uncertainty (80/20 contingencies) and periods of high uncertainty (60/40 contingencies). Participants were explicitly instructed that sometimes a token is fake and won't win any point, while sometimes it is valuable; and that the proportion of valuable tokens is opposite between the two tokens at any point in time, such that if 80% of orange tokens are valuable, only 20% of blue tokens are valuable. While participants did not know the contingencies, they were instructed that the partner knew the reward probability associated with each token. Participants were also told that the proportion of valuable tokens was the same for both players (such that what is inferred from the partner can be applied to the self), and that this proportion would switch many times during the task. In practice reversals occurred once per block, as well as in-between blocks. This design resulted in OL uncertainty blocks that lasted longer than EL uncertainty blocks, which was done to account for the possibility that OL would be slower than EL (given increased cognitive demands on the social inference process required in OL). Finally, reward magnitude was varied across blocks, such that in some blocks, rewarded token magnitudes would range between 45 and 55 points and in other blocks between 1 and 99 points.

### Procedure

Participants were first asked to fill out demographic questions at the beginning of the study, including age, gender, race, ethnicity and highest level of education. Then, the instructions and practice for the task contained two phases. Participants were first instructed about the tokens and the boxes and played a short practice (12 trials) which

helped them see the task from the perspective of the partner. Specifically, during this practice, they were explicitly told the proportion of valuable tokens before the start of each trial (i.e. they knew which token was more valuable), then had to choose between the two boxes to find which one was more likely to yield the more valuable token. They were then instructed that for the main task, they would not be told the percentage of valuable tokens anymore but would be able to observe the choices of another player who possessed that information. They were also informed that: "the other player is another participant who played a longer version of the practice you just played, and now we are replaying this person's action back to you". In practice the partner's actions were generated from a simple reinforcement learning model which updates the value of each box according to its history of yielding the more valuable token, and with an inverse temperature parameter of 10 (high choice consistency) and learning rate of 0.8. In total, 16 different trial lists were generated, and one was selected at random for each participant. A demonstration of the instructions, practice, and 10 example trials of the task can be found at this link: https://obsexplearn.web.app/.

At the end of the task, the following questionnaires were collected from all participants to examine individual differences in cognitive ability, as well as mood, anxiety and social traits. First, we collected a measure of IQ through the International Cognitive Ability Resource[67] (ICAR, https://icar-project.com/), a 16-item multiple-choice questionnaire assessing cognitive ability along four domains: letter and number series, verbal reasoning, three-dimensional rotation, and matrix reasoning. Participants then completed the State and Trait Anxiety Inventory[68] (STAI), the Beck Depression Inventory[69] (BDI-II), the Social Responsiveness Scale[70] (SRS-2), and the Liebowitz Social Anxiety Scale[71] (LSAS-SR). To ensure participants paid attention, there were three catch questions spread throughout the questionnaires: one attention question in the BDI ("If you are paying attention, select the last option"), one question to leave blank in the STAI, and one infrequency item in the SRS ("There are fifteen months in a year"). The task and questionnaires were coded up into a HTML URL, using custom Javascript code and plugins from jspsych versions 6.0.5. and 6.1.0.

## Behavioral analysis

Data was analyzed using Matlab (R2020b) and R (version 4.1.2). As an index of learning behavior, we computed the mean accuracy (i.e. propensity to choose the more valuable token) across individuals for the first 8 trials following a reversal in token value. If participants are learning, accuracy should increase over the course of those trials (Fig. 2A, B). We then focused on calculating model-agnostic behavioral signatures of EL and OL. To do that, we classified each trial as (1) consistent with EL or not and (2) consistent with OL or not, using the following definitions. A trial was deemed consistent with EL (Fig. 1C) if the participant chose the token that was rewarded on the previous trial or avoided the token that was previously unrewarded. A trial was deemed consistent with OL (Fig. 1D) if the participant chose the token obtained by the partner on the previous trial when the partner repeated their choice (e.g. the partner chooses Box A, gets an orange token, repeats choice of Box A, then participant chooses orange); or if the participant avoided the token obtained by partner on the previous trial when the partner changed their choice (e.g. the partner chooses Box A, gets an orange token, switches to Box B, then participants chooses blue). We calculated a behavioral index of the proportion of choices consistent with OL (versus EL) out of all the trials where EL and OL predicted different choices, according to the classification described above. This measure (Fig. 2C-D) represents individual preferences for OL versus EL in this task. To quantify whether participants combine the two strategies behaviorally, we ran a Mixed-Effects General Linear Model (ME-GLM) predicting choice on each trial from two predictors: the previous trial outcome (EL effect) and the partner's last action (OL effect) (Fig. 2E, F, Table S1A), with participant (*sub*) as a grouping

variable. The ME-GLM equation was as follows:

$$choice \sim 1 + out + pa + (1 + out + pa|sub) \quad (1)$$

where
$$choice = \begin{cases} 1 \ if \ orange \ token \ is \ chosen \\ 0 \ if \ blue \ token \ is \ chosen \end{cases},$$

$$out = \begin{cases} outcome(t-1) \ if \ last \ token \ was \ orange \\ -outcome(t-1) \ if \ last \ token \ was \ blue \end{cases},$$

and $$pa = \begin{cases} 1 \ if \ partner \ action \ is \ consistent \ with \ orange \ goal \ token \\ -1 \ if \ partner \ action \ is \ consistent \ with \ blue \ goal \ token \end{cases}$$

Note that *out* and *pa* regressors were only weakly correlated (Study 1: R(125) = 0.262; Study 2: R(493) = 0.272), indicating low shared variance (i.e. around 7 to 8%).

Finally, to examine arbitration behaviorally (i.e. the extent to which the preferred strategy changes with OL uncertainty, EL uncertainty and magnitude), we computed the breakdown in OL choice propensity according to each trial type, focusing on EL and OL uncertainty *trials* only (Fig. 3A, B), or further breaking down by magnitude (Fig. S2A, B). OL uncertainty was considered low on trials for which the past two partner's box-to-token transitions were consistent, and high otherwise. EL uncertainty was considered low on trials for which the previous two choices and outcomes followed what would be expected from a win-stay-lose-shift strategy (i.e., win-stay-win, win-shift-loss, loss-stay-loss, loss-shift-win), and high otherwise. Finally, reward magnitude was considered high if the last outcome magnitude was greater than 25 points, and low otherwise. High and low trials for each of the three variables are depicted by dots on Fig. 1B. We also ran a series of ME-GLMs to assess the effect of each uncertainty trial type on both EL and OL effects (see Fig. 3C, D for effects of EL and OL uncertainty trial types, Fig. S2C, D for effects of magnitude, and Table S1B–D for all ME-GLMs results). Specifically, each ME-GLM included the effect of past outcome (EL effect) and of the partner's last action (OL effect) separately for low and high uncertainty or magnitude trials. The ME-GLM equations were as follows, where *pa* represent the past partner action regressor, *out* the past outcome regressor, both as defined above, and *sub* is the grouping variable:

- Effect of OL uncertainty (Fig. 3C, D left, Table S1B):

$$choice \sim 1 + out_{low}^{OLu} + out_{high}^{OLu} + pa_{low}^{OLu} + pa_{high}^{OLu} \\ + \left(1 + out_{low}^{OLu} + out_{high}^{OLu} + pa_{low}^{OLu} + pa_{high}^{OLu}|sub\right) \quad (2)$$

where
$$out_{low}^{OLu} = \begin{cases} out \ if \ OL \ uncertainty \ is \ low \\ 0 \ if \ OL \ uncertainty \ is \ high \end{cases},$$

$$out_{high}^{OLu} = \begin{cases} 0 \ if \ OL \ uncertainty \ is \ low \\ out \ if \ OL \ uncertainty \ is \ high \end{cases},$$

$$pa_{low}^{OLu} = \begin{cases} pa \ if \ OL \ uncertainty \ is \ low \\ 0 \ if \ OL \ uncertainty \ is \ high \end{cases},$$

$$pa_{high}^{OLu} = \begin{cases} 0 \ if \ OL \ uncertainty \ is \ low \\ pa \ if \ OL \ uncertainty \ is \ high \end{cases}$$

- Effect of EL uncertainty (Fig. 3C, D right, Table S1C):

$$choice \sim 1 + out_{low}^{ELu} + out_{high}^{ELu} + pa_{low}^{ELu} + pa_{high}^{ELu} \\ + \left(1 + out_{low}^{ELu} + out_{high}^{ELu} + pa_{low}^{ELu} + pa_{high}^{ELu}|sub\right) \quad (3)$$

where
$$out_{low}^{ELu} = \begin{cases} out \ if \ EL \ uncertainty \ is \ low \\ 0 \ \ if \ EL \ uncertainty \ is \ high \end{cases},$$

$$out_{high}^{ELu} = \begin{cases} 0 \ if \ EL \ uncertainty \ is \ low \\ out \ if \ EL \ uncertainty \ is \ high \end{cases},$$

$$pa_{low}^{ELu} = \begin{cases} pa \ if \ EL \ uncertainty \ is \ low \\ 0 \ if \ EL \ uncertainty \ is \ high \end{cases},$$

$$pa_{high}^{ELu} = \begin{cases} 0 \ if \ EL \ uncertainty \ is \ low \\ pa \ if \ EL \ uncertainty \ is \ high \end{cases}$$

- Effect of reward magnitude (Fig. S2C, D, Table S1D):

$$choice \sim 1 + out_{low}^{Mag} + out_{high}^{Mag} + pa_{low}^{Mag} + pa_{high}^{Mag}$$
$$+ \left(1 + out_{low}^{Mag} + out_{high}^{Mag} + pa_{low}^{Mag} + pa_{high}^{Mag} \mid sub\right) \qquad (4)$$

where
$$out_{low}^{Mag} = \begin{cases} out \ if \ reward \ magnitude \ is \ low \\ 0 \ if \ reward \ magnitude \ is \ high \end{cases},$$

$$out_{high}^{Mag} = \begin{cases} 0 \ if \ reward \ magnitude \ is \ low \\ out \ if \ reward \ magnitude \ is \ high \end{cases},$$

$$pa_{low}^{Mag} = \begin{cases} pa \ if \ reward \ magnitude \ is \ low \\ 0 \ if \ reward \ magnitude \ is \ high \end{cases},$$

$$pa_{high}^{Mag} = \begin{cases} 0 \ if \ reward \ magnitude \ is \ low \\ pa \ if \ reward \ magnitude \ is \ high \end{cases}$$

All trial-by-trial ME-GLMs predicting choice (Table S1) were run using the *fitglme* function in MATLAB and included both fixed and random effects of each predictor, as well as a fixed and random intercepts. Effect sizes were calculated as Cohen's *d* for t-tests (two-tailed), and partial eta-squared $\eta_p^2$ for F-tests. Normality was not formally tested, and equal variances were not assumed for t-tests. 95% confidence interval (CI) are reported wherever possible. To provide evidence for the null effect in the case of t-tests, Bayes Factors ($BF_{10}$) were calculated using *R*'s *BayesFactor* package[72].

## Computational models of individual strategies

A set of 5 models representing different strategies were defined and fit to the data. The rationale for this approach was to identify which model is more likely to be used by each participant, in order to characterize the heterogeneity in strategy use.

- Experiential learning (EL) model

EL was modelled as simple reinforcement learning of the reward probability associated with each token, combined with a magnitude boosting mechanism. Token values were initialized at 0.5 for each token. The value of the chosen token was then updated, with learning rate $\alpha_{exp}$ and experiential reward prediction error *eRPE*:

$$TokV_{exp,ch}(t) = TokV_{exp,ch}(t-1) + \alpha_{exp}*eRPE(t) \qquad (5)$$

where $eRPE(t) = \begin{cases} 1 - TokV_{exp,ch}(t-1) \ if \ reward \\ 0 - TokV_{exp,ch}(t-1) \ if \ no \ reward \end{cases}$

The value of the unchosen token was inferred given the knowledge that the reward probabilities of both tokens sum to 1:

$$TokV_{exp,unch}(t) = 1 - TokV_{exp,ch}(t) \qquad (6)$$

In parallel, the reward magnitude *M* associated with each token was tracked such that if a token was rewarded, *M* was updated to the reward value, whereas if a token was unrewarded, or unchosen, *M* was decayed by 50% relative to its value on the previous trial. This was implemented following behavioral analyses showing how previous reward magnitude enhanced learning and following modelling analyses of Study 1 data to establish the best-fitting EL mechanism (see Table S3 for details). The probability of choosing orange was then obtained through a softmax of the value difference between the orange (*or*) and the blue (*bl*) tokens:

$$P_{exp}^{or}(t) = \frac{1}{1 + e^{\left(-\beta_{exp}*\left(TokV_{exp}^{or}(t) - TokV_{exp}^{bl}(t)\right) - \mu*\left(M^{or}(t) - M^{bl}(t)\right)\right)}} \qquad (7)$$

where $\beta_{exp}$ is the inverse temperature parameter and $\mu$ represents the magnitude boosting effect, i.e. the extent to which the probability of choosing the orange (vs blue) token is influenced

by the past magnitude of rewards obtained with the orange (vs blue) token.

- Observational learning (OL) model

OL was modelled through reinforcement learning of the transition probabilities between the partner's choices and tokens. The probability that the action performed by the partner leads to an orange (vs blue) token was updated with learning rate $\alpha_{obs}$ and observational state prediction error *oSPE*:

$$ActV_{obs,ch}(t) = ActV_{obs,ch}(t-1) + \alpha_{obs}*oSPE(t) \qquad (8)$$

where $oSPE(t) = \begin{cases} 1 - ActV_{obs}(t-1) \ if \ orange \ token \\ 0 - ActV_{obs}(t-1) \ if \ blue \ token \end{cases}$

A counterfactual update of the value of the unchosen action was also added, since instructions specified that one box had a higher proportion of orange tokens and the other one a higher proportion of blue tokens:

$$ActV_{obs,unch}(t) = ActV_{obs,unch}(t-1) - \alpha_{obs}*oSPE(t) \qquad (9)$$

The partner's goal (i.e., token values) were then directly inferred from the action values, such that if the partner's action has a 70% chance of leading to an orange token, the assumption is that the orange token has a 70% chance of being more valuable:

$$TokV_{obs}(t) = [ActV_{obs}(t), 1 - ActV_{obs}(t)] \qquad (10)$$

Choice probability was then calculated as a softmax function of the token value difference, with inverse temperature parameter $\beta_{obs}$:

$$P_{obs}^{or}(t) = \frac{1}{1 + e^{\left(-\beta_{obs}*\left(TokV_{obs}^{or}(t) - TokV_{obs}^{bl}(t)\right)\right)}} \qquad (11)$$

- Fixed mixture model

In this model, choice probabilities predicted by OL ($P_{obs}^{or}$) and by EL ($P_{exp}^{or}$) were combined using a fixed weight parameter $\omega_{OL>EL}$, which represents the probability of relying on OL over EL.

$$P^{or}(t) = \omega_{OL>EL}*P_{obs}^{or}(t) + (1 - \omega_{OL>EL})*P_{exp}^{or}(t) \qquad (12)$$

- Dynamic arbitration model

In this model, $\omega_{OL>EL}$ was no longer a free parameter estimated for each participant but varied dynamically depending on the reliability of each strategy. Unsigned prediction errors were used as an index of how unreliable each strategy was, consistent with the hypothesis that when a strategy is reliable it should generate small prediction errors. Specifically, the reliability of OL depended on the min-max normalized observational state prediction error (scaled between −1 and +1):

$$R_{OL}(t) = -\left(2*\frac{|oSPE(t)| - \min(|oSPE|)}{\max(|oSPE|) - \min(|oSPE|)} - 1\right) \qquad (13)$$

This means that when the state transitions between the partner's actions and token were predictable (small prediction errors), the reliability of OL was high, as inferring the goal token from observing the partner's actions was easier.

The reliability of EL (also scaling from −1 to +1) depended on the min-max normalized experiential reward prediction error and on the scaled outcome magnitude from the previous trial:

$$R_{EL}(t) = -\frac{|eRPE(t-1)| - \min(|eRPE|)}{\max(|eRPE|) - \min(|eRPE|)} + \frac{outcome(t-1)}{100} \qquad (14)$$

EL was therefore most reliable when outcomes associated with the chosen token were predictable (small prediction errors) and higher in magnitude.

The arbitration weight was then calculated as a softmax of the reliability difference between the two strategies, as well as a bias parameter $\delta_{OL>EL}$ capturing the preference for OL over EL:

$$\omega_{OL>EL}(t) = \frac{1}{1 + e^{-(R_{OL}(t) - R_{EL}(t) + \delta_{OL>EL})}} \tag{15}$$

This dynamic weight could then be used to combine the two strategies and calculate the choice probability, in a similar fashion as the fixed mixture model:

$$P^{or}(t) = \omega_{OL>EL}(t) * P^{or}_{obs}(t) + (1 - \omega_{OL>EL}(t)) * P^{or}_{exp}(t) \tag{16}$$

- Baseline strategies model

This model was added to the set to capture the behavior of 'non-learner' participants who did not rely on EL or OL to learn and instead used an irrelevant (but non-random) strategy. Specifically, four strategies were included in that model, captured by four separate parameters: a color bias (preference for orange over blue token), a left-right bias (preference for left over right action), a sticky action bias (tendency to repeat the past left or right action), and an action imitation bias (tendency to repeat the partner's left or right action).

## Model fitting

Model fitting was done with the computational and behavioral modeling (*cbm*) toolbox[51] in Matlab (R2020b), using Laplace approximation with a normal prior for each parameter with mean 0 and variance 6.25. First, to ensure that the 5 models could be appropriately dissociated, a confusion matrix was calculated by simulating data ($N = 100$ simulated datasets) from each of the models and fitting the simulated data using each model, then calculating the exceedance probability through *cbm* toolbox's hierarchical Bayesian fitting (i.e. which model in the set can explain the simulated datasets best; Fig. S3A). We then performed parameter recovery for each of the five models. To do so we generated 10 different datasets for each model and each participant, using their best-fitting parameter estimates, then re-fit the model to the generated data, correlated the actual parameters used to generate data with the recovered parameters and averaged the correlation coefficients across the 10 iterations (Fig. S3B–F). All models were fit to each participant's data, first computing individual-level fits followed by hierarchical Bayesian fitting to obtain more reliable parameter estimates. We report the following three comparison metrics for all models (Table 1): AIC, out-of-sample predictive accuracy (which was calculated by leaving one block out, fitting the model on 7 blocks out of 8, and predicting choice on the remaining block, then repeating and averaging across blocks, see also Fig. 5A, B), and model frequencies from the hierarchical fitting.

## Posterior predictive checks

Finally, we performed several posterior predictive check analyses to ensure the validity of each strategy's behavioral signature, of the five groups, and of the dynamic arbitration scheme. First, choice data was generated from each participant's best-fit parameters for the OL and EL models, then the proportion of those choices consistent with OL (vs EL) behavior was calculated for each generated choice set, averaged across 1000 simulations, and compared with the participants data (Fig. S4). Second, we generated choice data for all five models (also from each participant's best-fit parameters) and ran the ME-GLMs (shown in Fig. 2E, F) that estimate the main effects of past outcome (EL effect) and past partner's action (OL effect) separately on each model's generated choices. We report the main effects for the data next to the model predictions (Fig. 4A, B) as well as correlations between the data and model predictions across individuals for the EL, OL and dynamic arbitration models (Fig. 4C, D). Third, to ensure that the dynamic arbitration model could appropriately capture the effect of uncertainty on each strategy, we ran the ME-GLMs (shown in Fig. 3C, D) that estimate the EL and OL effects separately for low and high EL and OL trial uncertainty on data generated by the dynamic arbitration model and show the resulting ME-GLMs individual estimates (random effects) next to those estimated from the data (Fig. 4E–H). The correlations between ME-GLM random effect estimates from participants data and from model-generated data for this analysis were also computed, separately for each effect and each uncertainty type (Fig. S5). Finally, we extracted the trial-by-trial values of the arbitration weight ($\omega_{OL>EL}(t)$) from the dynamic arbitration model, separately for each uncertainty/magnitude trial types, to ensure that the model-predicted weight values varied as expected with OL uncertainty, EL uncertainty and reward magnitude (Fig. S6).

## Group differences in behavior

We classified participants into five groups depending on which model fit each participant best using individual model frequencies output by hierarchical model-fitting as the metric for model comparison at the individual level. We then calculated and plotted separately for each of the five groups the different behavioral metrics of interest detailed above (Behavioral analysis section): learning curves (Fig. 5D, E), EL and OL ME-GLM effects (Fig. 6A, B), behavioral index of arbitration (Fig. 7A, B), as well as the effects of EL uncertainty trials (Fig. 7D, F), OL uncertainty trials (Fig. 7C, E), and magnitude (Fig. S2E, F) on strategy use. These variables were averaged separately for each group, and the effect of group was assessed using regression analyses in R (*lme4* package), including a random intercept and controlling for gender, age, education level, and cognitive ability (ICAR score), and followed by type III analysis of variance (see Table S4 for equations and statistics). We also performed some of the posterior predictive check analyses broken down by group to ensure that each group's model (i.e. EL for ExpLearn group, OL for ObsLearn group, and so on) can appropriately predict behavior in that group, both by using the actual groups defined by the model frequencies as well as randomly shuffled groups (where each participant is assigned to a group at random). We report this analysis for the learning curves (Fig. 5F–I) and for the ME-GLM EL and OL effects (Fig. 6C–F).

## Careless exclusion

Before examining associations between behavioral tendencies on the tasks and questionnaire scores, we excluded additional participants who showed evidence of careless responding on the questionnaires, similar to the approach used in a recent paper[73], and recently recommended as a way to avoid spurious possible associations with task performance[57]. Specifically, we first excluded participants who failed to correctly answer one or more catch questions and participants who missed one or more questionnaires (data lost or study stopped before the end). Second, we ran the R package *careless*[74] to assess random and inattentive responding on the questionnaires (excluding ICAR which is a separate measure of cognitive ability). We computed the intra-individual response variability on the SRS and the STAI (both contain reverse-coded questions, so responses should show some variability), as well as even-odd consistency, psychometric synonym and antonym scores averaged across all the questionnaires. Those measures were z-scored, and participants excluded if any of these scores was more than two standard deviations from the mean in the 'unwanted' direction (i.e. $Z < -2$ for intra-individual response variability, even-odd consistency and psychometric synonym, and $Z > 2$ for psychometric antonym). In total, out of 619 participants from both studies, 51 participants were excluded, leaving a final pooled sample size of $N = 568$. Note that

performance on the task was not different between these later excluded participants and all remaining participants, hence why they were only excluded at this stage of the analysis.

## Questionnaire factor analysis

To extract meaningful transdiagnostic symptom dimensions from the questionnaire scores, we performed exploratory factor analysis using the R package *fa*. Because of the exploratory nature of this analysis, we pooled data from the two studies, thus performing the factor analysis on the final pooled sample of 568 participants. Following some recommendations on exploratory factor analysis[75], we used the Weighted-Least Square fitting method combined with *oblimin* factor rotation. To determine the optimal number of factors, we ran the factor analysis with 1 to 20 factors and extracted BIC as a goodness-of-fit criterion (Fig. S9A). We assessed the significance of group differences in factor scores using *lme4*, modelling score as a function of a factor-by-group interaction, controlling for study, gender, age, education level, and ICAR score, and including a random intercept (Fig. 8, Table S5).

## Reporting summary

Further information on research design is available in the Nature Portfolio Reporting Summary linked to this article.

## Data availability

The raw (trial-by-trial) and summary (participant-level) data generated in this study are available at: https://github.com/ccharpen/OL_EL_behavior; and https://doi.org/10.5281/zenodo.10695037[76].

## Code availability

All code used in this study to run the experiment online, analyze the data and generate the figures, tables and results reported in this manuscript is available on the following repository: https://github.com/ccharpen/OL_EL_behavior; and https://doi.org/10.5281/zenodo.10695037[76].

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

## Acknowledgements

This work was funded by the NIMH Caltech Conte Center grant on the neurobiology of social decision-making (P50MH094258) to J.P.O.D., as well as a Wellcome Trust Sir Henry Wellcome Postdoctoral Fellowship (218642/Z/19/Z) and NIMH K99/R00 award (K99MH123669) to C.J.C.

## Author contributions

C.J.C. and J.P.O.D. conceptualized the study. C.J.C. designed the experiment, collected and curated data. C.J.C. conceptualized the computational models with support from Q.W., W.D., J.C., and J.P.O.D. C.J.C. analyzed the data, with support from Q.W. and S.M. C.J.C. wrote the original draft of the manuscript, with support from J.P.O.D. All authors reviewed and edited the subsequent versions of the manuscript. J.P.O.D. supervised the work, with support from C.J.C. and J.C. J.P.O.D. and C.J.C. acquired funding.

## Competing interests

The authors declare no competing interests.
