## [Peer Review File · Nature Communications]

Heterogeneity in strategy use during arbitration between experiential and observational learningREVIEWER COMMENTS

Reviewer #1 (Remarks to the Author):

The work by Charpentier et al. investigates arbitration strategies between experiential and observational learning as well as heterogeneity in various employed strategies across an unselected population with varying psychopathology traits. This work is innovative and methodologically rigorous, using novel tasks and modeling methods to parse relevant strategies. The justification of an unselected online sample is appropriate and the authors make a good case for its utility in this context. The paper is well presented and motivated, and the results are interesting and timely for cognitive neuroscience and computational psychiatry.

In general, I am very enthusiastic about this work and have only relatively minor concerns:

Main points:

1. Model comparison results.

1a. I find some aspects of the model comparison results somewhat puzzling. First, unless I am misreading or missing something, it is a bit surprising that the exceedance probability for the fixed mixture model is <0.001 for study 1 and 0.950 for study 2. In other work the results from study 2 could have been taken at support for the fixed mixture model as the winning model. The authors highlight that the model probabilities are all relatively low and roughly evenly distributed, but it would be good to discuss and justify this further in the context of the previous literature.

1b. Second, the ME-GLM regression weights for the DynArb model are stronger than those for the ObsLearn and the ExpLearn models for both EL and OL effects. This is surprising to me because I would have thought that the DynArb model would have a less consistent (and presumably weaker) EL effect than the ExpLearn model and a less consistent (and presumably weaker) OL effect than the ObsLearn model. The authors argue that this is explained by higher accuracy in the DynArb group, but I still wonder if this would be expected by the model and hold in posterior checks (i.e., if data generated by the best fitting models and parameters for each group subjected to the ME-GLM would recapitulate this pattern of OL and EL weights). I also wonder whether this could be consistent with the relative model evidence being weaker for ExpLearn and ObsLearn groups compared to the DynArb group. Could the authors provide any additional data to address this, for instance absolute goodness-of-fit metrics (eg, pseudo-R²) and differential model comparison metrics (eg, delta BIC/AIC) for each model/group? What would be the implications if the DynArb group are not only using a unique strategy but are better learners overall (i.e., they differ both qualitatively and quantitatively).

2. I was surprised to see the details of the EL model with regard to the incorporation of reward magnitude. Instead of being incorporated into the EV calculation, the magnitude is learned separately and affects choice directly. The authors mentioned that the magnitude effects were unexpected and they seem to have modified the model accordingly, but it is still surprising that this reward magnitude effect is not integrated into the token EV, and is learned separately only biasing choice. I would like to see more justification and model comparison to for this model architecture, as well as discussion on how this may affected the overall model comparison of different strategies.

Minor points:

- Toning down language on psychiatric illness and comorbidity. While the authors are generally cautious about terminology, in a few of places they talk about benefits regarding "comorbid symptom dimensions that may be confounded in the clinic" (line 2, p. 17), "psychiatric profiles" (line 22, p. 19), "symptom dimensions that are typically comorbid in the general population" (line 30, p. 20). I would recommend avoiding the terms comorbidity and psychiatric, because they imply pathological illness.

The traits studied here may or may not be pathological. Also, autism spectrum disorder and anxiety disorders are not usually confounded in the clinic in that there is no difficult differential diagnosis that is required. So they are comorbid but to my knowledge there is no actual clinical need to deconfound them. In general, and while I do see the value of this work for computational psychiatry, I think it is more appropriate to use terms like dimensional psychopathology traits/symptoms, and coexisting symptoms/traits, and tone down direct implications of this work for clinical psychiatry (given the need to confirm potential relevance in clinical populations).

- In the discussion, the authors argue for the benefit of including poor learners and that poor performance may be indicative of pathologies of interest. They argue that broad exclusion could lead to a reduction in the range of relevant symptom dimensions associated with the use of non-learner strategies. While in general I agree with their approach and see its benefits, absent from this discussion is the limited interpretability of non-learning strategies. In certain psychopathologies (eg, a severe motivational impairment in schizophrenia or MDD), one may see poor performance and learning that is somewhat generic. It may simply represent poor motivation to engage with the task versus an actual strategy or diminished capacity to perform the task using a more appropriate strategy. In turn, impairments in higher-order cognitive function or memory could lead to worse task construal or rule maintenance, which could end up resulting in a similar phenotype. So poor learning may be a mixed bag with various potential interpretations (and unexplored potential mechanisms), and in itself it may not be informative of well-specified computational failure modes. Therefore, the key advantage of interpretability enabled by theory-driven modeling approaches is in my mind less clearly applicable to this particular group. Perhaps it would be good to provide a more balanced discussion on this point to briefly represent this counterargument.

- Typo: "different point in times" (line 17, p. 2)

Reviewer #2 (Remarks to the Author):

In this clear and detailed paper, the authors report that individuals use a combination of strategies - experiential and observational learning - in order to maximise rewards and minimise punishments. Importantly, they show that there are subgroups of individuals who use different combinations of these strategies - with some showing their hypothesised 'dynamic arbitration' pattern, in which individuals choose between these strategies depending on the prediction errors received. I think this is a novel paper that addresses an important question, but I have some theoretical and some writing comments.

My main theoretical comment is that the two learning strategies actually form different parts of the task - the early part of each trial involves another player making choices, and the individuals having to infer from their choices and the instructions what the other player is intending. This is complex, and relies on many cognitive processes (working memory to recall the instructions and practice to understand what the *****other***** player knows; theory of mind, etc) - other than just experiential learning. Additionally, the player must assume that the other player is learning - another process to account for when making inference about their choices. More comment on this in the discussion might be warranted, as I do not think that the task equates the two 'learning' processes exactly (either in terms of difficulty or confounds).

Relatedly, here the process of 'dynamic arbitration' is dependent only on the environment, and the environment's uncertainty at that: in many other arbitration processes (e.g. model-based/model-free learning) arbitrating between strategies relies also on the individual's goals, or other environmental features. This is briefly mentioned in the introduction (end of page 2), and could be expanded on.

Some more minor points:

*****Abstract*****

In the abstract, 'control over behaviour' is assigned to learning strategies. This wording seems somewhat awkward - are these learning processes really also acting as controllers?

I felt that the start of the introduction was framed heavily around social skills, which are important but I don't believe experiential learning only relates to social skills.

Typo: Should be explosion ***of*** online data collection, I think (line 6, page 3)

The link between online testing and different strategies seemed a little forced - surely people use different strategies in person too? Maybe this point could be made clearer?

Results

Uncertainty is defined in quite a basic way, based on the last few trials. Of course modelling is used eventually, but even in this case uncertainty is defined based on prediction errors. Maybe this could be commented on in the discussion - more sophisticated estimates of uncertainty, based on more than just a couple of trials back, could have been used.

Some figures seem to have boxes around them, including in the supplement, which should be fixed if possible.

In the ME-GLM, I wasn't sure whether significant effects could be seen from trials where EL and OL would produce a consistent result - i.e., are these effects really specific to EL and OL? In the first analyses only trials that are consistent with one and not the other are used, presumably to avoid this issue.

I disagree with the use of 'marginal' to describe non-significant p-values - these should not be reported (pages 15-16).

Typo: I think it should say 'trials' not 'trial' (line 2, page 24)

Could the authors comment on whether there is enough power to draw strong conclusions about the differences between groups? In particular, some groups only have a very small number of individuals within them.

Discussion

If reward magnitude effects are only in the supplement, they shouldn't be discussed in main text - either this discussion should be in the supplement, or the results of the reward magnitude analyses should be presented in the main text.

I agreed with the points made re: baseline models and including data/relevance for psychopathology.

Methods

Was the exclusion criteria (25% of trials) pre-planned? It also isn't clear from the methods that participants were also excluded for missing catch questions - I would perhaps appreciate a sensitivity analysis to check the exclusion of these participants aren't affecting these results.

What does deemed exempt mean with regards to ethical approval?

The choice of 50% decay based on 'previous analyses' isn't well-justified - perhaps this could be

expanded on in the Supplement.

The data and code statement is not up to date in the main text - this is already available I think.

Supplement

For parameter recovery, it would be clearer if white was 0, and red/blue were the opposite ends of the spectrum - strong negative correlations between parameters should not appear white!

Figure S6a looks like 7 factors might have a lower BIC? Perhaps presenting the BIC numbers alongside would be helpful here.

Reviewer #3 (Remarks to the Author):

In this paper, Charpentier and colleagues investigate the arbitration between experiential and observational learning in 2 samples of human participants. They designed a task in which participants have to select one of 2 colored tokens. One of the colored tokens is (probabilistically) more rewarded, and this changes over time (actually once per block of 20 trial). Participants can figure out how to optimize their behavior from 1) their past history of choices and rewards (experiential), and 2) observing another participant selecting boxes probabilistically associated with token (observational). Because this other participant knows which token is most rewarded, the combination of their choices and past outcomes can be informative about the identity of the best token.

Overall, the topic is interesting, and the paper features some high-quality characteristics, such as the large number of participants and the presence of a replication sample.

I nonetheless have serious concern about the framing of the study, the modelling and the robustness of the conclusions, which make me doubt the appropriateness of the current version of this manuscript for Nature Communications. I hope my comments will be helpful to the authors to improve this manuscript, which builds on a potentially interesting setup.

1. First and most critically, in some instances, the analytical strategies which support the current story do not seem to fit with the actual experimental design, making me quite uncomfortable with the whole piece. From what I understand, the actual experimental design is a repetition of blocks of 20 trials (each with new pairs of fractals) featuring a fixed OL uncertainty level and one reversal in the EL contingencies. It is definitely not a symmetrical manipulation of OL and EL levels of uncertainty (changes in levels of OL uncertainty are clearly announced by a change of cues; changes in levels of EL uncertainty have to be detected by trials-and-errors), contrary to what the manuscript seems to suggest with the "arbitration" story. Similarly, changes in OL uncertainty cannot be conceptualized as "switches", because each pair of fractals have a fixed box-to-token transition probability (though it is not clear if, e.g. Fig 4 & C consider as "switches" only the proper reversal in EL, or all changes in EL and OL contingencies). In the end, for most behavioral analyses, the authors rely on a very arbitrarily defined notion of uncertainty for EL and OL [p5: EL uncertainty (blue points) was deemed low on trials where past outcome-action-outcome sequence was consistent, and high otherwise. OL uncertainty (orange points) was deemed low if the past two box-token transitions were consistent, and high otherwise], and the writing seem to suggest that those have been experimentally manipulated over the course of the experiments. I would certainly have more faith in a set of analysis that explore the role of properly manipulated factors (i.e. actual levels of probability) rather than of those arbitrary metrics. Those concern may seem trivial, but I think they actually jeopardize a significant share of the narrative that the proposed analyses are meant to support.

2. Second, though apparently sophisticated, I found the whole modelling section uncharacteristically weakly supported by the data. At no point do the author show model fits (i.e. model predictions/fits

against actual data, e.g. learning curves), replications of behavioral effects/analyses with model fits (e.g. the GLME featured in Fig 2-3, but with model simulations), or interpretable indexes of model goodness of fit (or absence thereof). Only Figure S3 provides some results on posterior predictive checks but on such an aggregated level that it is hard to feel confident that the model even remotely fit and reproduce participants behavioral data when it comes to the effects of experimental manipulations. I would strongly suggest that the authors 1) show model fits supported by data when appropriate (e.g. learning curves, etc.) 2) provide model simulations that show that synthetic data can replicate all the effects of the experimentally manipulated factors on behavior, as shown on the GLME on behavior (effects of high vs low EL vs OL uncertainty + magnitude), 3) provide estimate of goodness of fit, such as average (out-of-sample) likelihood per trial (of note, these model quality-check analyses seem to have extensively been performed in a previous manuscript of the same team (Charpentier, et al 2020 Neuron)).

3. Other modelling various modelling issues. Like the behavioral analyses, the model design seems to overly rely on somewhat arbitrary decisions (like the magnitude mechanism, or the computation of EL and OL reliability). At the very minimum, I would like to see some analyses that the chosen implementations actually succeed in capturing the said mechanisms/effects. Regarding parameter recovery, in my understanding, averaging estimated parameters over 10 simulations before correlating them with the generative ones leads to grossly overestimating recovery, because this averages out the noise inherent in the estimation – hence does not give a fair estimate of the reliability of the parameters estimated in the actual dataset. A probably more fair approach would be to compute the correlation matrix in each of the 10 simulation independently, and then compute an averaged correlation matrix over simulations.

4. Although I appreciate the attempt to tackle the heterogeneity of individual strategies, I have a feeling that in the present case, these results are more a symptom that the task (and/or models) does not capture a reliable process. In my understanding, the same results (all models seem represented in the data and there is no clear winner) would be obtained if no model is good at capturing any of the data (rather than the proposed interpretation that all models are somewhat good at capturing different share of the data). The “external validation” through the trans-psychiatric-like analysis is not overwhelmingly convincing (statistically speaking). Here, we come back to the necessity of obtaining clear and convincing assessment of the goodness-of-fit of the models (see my point 2).

5. Finally, I have some doubts about the conceptual validity of the experiment as an instance of “observational” learning. Arguably, the cover story of the task is pretty weak: the (fake) partner “making choices” presumably know the reward probability associated with each token; i.e. including the (un-signalized) reversals. This seems hardly believable, in which can wonder whether the task really investigates “observational” learning, or more credibly a hierarchical inference process akin to model-based/model-free types of dichotomies?

November 15th, 2023

Dear Reviewers,

Thank you for reviewing our manuscript, “*Heterogeneity in strategy use during arbitration between experiential and observational learning*” (NCOMMS-23-16334-T) and for your astute concerns and suggestions. We have taken all of them into account when revising the manuscript and believe the revised manuscript is now considerably strengthened as a result.

Specifically, we have added substantial analyses to strengthen and validate our modelling approach (as suggested by all Reviewers), including (i) extensive posterior predictive check analyses showing that our models can reproduce the expected behavioral patterns, both across all participants (**Figures 4 & S5**) and within groups (**Figures 5 & 6**), (ii) goodness-of-fit metrics (in particular out-of-sample accuracy, **Table 1 & Figure 5**), and (iii) further modelling analyses to justify the experiential learning magnitude boosting mechanism (**Table S3**). We have also addressed Reviewer #3’s concern about our definition of uncertainty by showing that similar patterns of findings are observed when using the uncertainty block conditions rather than the trial definition, while providing additional analyses to show that uncertainty trials are a better predictor of choice than uncertainty conditions (**Table S2**), in line with our hypothesis that the trial-based definition more closely characterizes the uncertainty experienced by the participant. As suggested by Reviewer #2, we have expanded our discussion section to comment on the processes involved in the two learning strategies as well as factors influencing arbitration, and have conducted post-hoc power analyses on group difference analyses. In line with Reviewer #1’s comments, we have toned down the language on psychiatric illness and comorbidity, and now discuss the limited interpretability of the non-learner group from a mechanistic standpoint.

For ease of reference, the changes are highlighted below, with the reviewers’ comments in bold followed by our response to each concern. Specifically, responses to Reviewer #1 start on **page 2** of this document, responses to Reviewer #2 on **page 11**, and responses to Reviewer #3 on **page 18**.

With those revisions, we hope the manuscript will now be deemed suitable for publication in *Nature Communications*.

Yours sincerely,

Caroline J. Charpentier & John P. O’Doherty

Reviewer #1:

The work by Charpentier et al. investigates arbitration strategies between experiential and observational learning as well as heterogeneity in various employed strategies across an unselected population with varying psychopathology traits. This work is innovative and methodologically rigorous, using novel tasks and modeling methods to parse relevant strategies. The justification of an unselected online sample is appropriate and the authors make a good case for its utility in this context. The paper is well presented and motivated, and the results are interesting and timely for cognitive neuroscience and computational psychiatry.

In general, I am very enthusiastic about this work and have only relatively minor concerns:

Main points:

1. Model comparison results.

1a. I find some aspects of the model comparison results somewhat puzzling. First, unless I am misreading or missing something, it is a bit surprising that the exceedance probability for the fixed mixture model is <0.001 for study 1 and 0.950 for study 2. In other work the results from study 2 could have been taken at support for the fixed mixture model as the winning model. The authors highlight that the model probabilities are all relatively low and roughly evenly distributed, but it would be good to discuss and justify this further in the context of the previous literature.

We thank the reviewer for the opportunity to clarify the model-fitting procedure. We were also surprised by the values indicated by the exceedance probability, especially given that all the other model-fitting metrics (AIC and out-of-sample predictive accuracy, which we have now added to **Table 1**, as well as model frequencies) suggest that the model fits are fairly similar across both studies. After exploring those metrics and the model-fitting procedures in more detail, we believe that exceedance probability is not a very reliable metric for model comparison in the context of heterogeneous samples given that the algorithm is trying to maximize finding a winning model and therefore treating the sample as homogeneous. Though exceedance probability is commonly reported in the literature and provides a convenient method for identifying a winning model to explain a given dataset, it appears that it is not a good measure to capture heterogeneity in strategy use across individuals. We found this to be the case across our two independent datasets. Further exploration of the literature also confirms that exceedance probabilities (EP) cannot be interpreted as a statistical significance threshold and do not directly represent how likely a model is relative to others, but rather the probability that any model is more frequent than the others. Specifically, Rigoux et al (2014, *Neuroimage*, 84: 971-985) explored the assumptions, usage and interpretation of EP, in terms of the “statistical risk incurred when performing random effects BMS” – that is allowing for heterogeneity across individuals. They conclude that “EP cannot be used to assess this statistical risk. More precisely, EPs are slightly overconfident — for example, if the best model has an EP of $\phi=0.95$, the probability that there is no difference in model frequencies is greater than 0.05. This is because the definition of EP does not consider a null model at the group level. In other words, chance is discounted as a potential explanation for the data. Although this does not invalidate EP-based ranking of candidate models, it means that one should not equate it with classical 5% significance thresholds”. In light of this evidence, we have now removed the exceedance probability metric and instead added AIC and out-of-sample predictive accuracy for all models, also consistent with metrics commonly reported in the literature. We now see that the fixed mixture model performs similarly in the two studies.

*Results, p.10: “Model frequencies from the latter analysis, as well as AIC and out-of-sample model predictive accuracy averaged across participants (see **Methods** for details) are reported in **Table 1**. Overall, those findings*

suggest that there was no clear and consistent winner. In both studies, the AIC values suggest a marginal advantage for the fixed mixture model, while out-of-sample accuracy slightly favored the dynamic arbitration model. Additionally, the model frequency values suggest somewhat of an even split, with no model exhibiting a frequency higher than 33%, with Study 1 showing the largest frequency for the observational learning model (31.1%) and Study 2 for the fixed mixture model (32.4%).”

Table 1, p.10:

Model	N _{par}	Study 1				Study 2			
		AIC	OOS acc	Frequency	N _{best} (% _{tot})	AIC	OOS acc	Frequency	N _{best} (% _{tot})
Baseline	4	215.7	0.521	0.205	25 (19.8)	219.8	0.511	0.159	83 (16.8)
Experiential learning	3	207.7	0.539	0.147	21 (16.7)	204.9	0.552	0.060	24 (4.9)
Observational learning	2	197.2	0.569	0.311	40 (31.8)	194.8	0.575	0.190	95 (19.3)
Fixed mixture	6	191.0	0.593	0.115	14 (11.1)	186.0	0.607	0.324	160 (32.5)
Dynamic arbitration	6	191.2	0.595	0.222	26 (20.6)	187.1	0.608	0.267	131 (26.6)

Table 1. Summary of model fits. Each of the five models (N_{par} = number of parameters) was fitted to participants data first using Matlab’s *cbm* toolbox. Using individual model-fitting, we computed the mean AIC as well as mean out-of-sample accuracy (OOS acc), which was calculated for each individual by fitting the model on 7 task blocks and using the best-fitting parameters to calculate the likelihood of predicting the participant’s choices in the remaining block (then iterating across all 8 blocks). We then using *cbm*’s hierarchical Bayesian inference fitting across all five models to compute model frequency. Selecting the best-fitting model for each individual participant (highest model responsibility), we then calculated the number and proportion of participants for whom each model explains their data best (N_{best} column).

1b. Second, the ME-GLM regression weights for the DynArb model are stronger than those for the ObsLearn and the ExpLearn models for both EL and OL effects. This is surprising to me because I would have thought that the DynArb model would have a less consistent (and presumably weaker) EL effect than the ExpLearn model and a less consistent (and presumably weaker) OL effect than the ObsLearn model. The authors argue that this is explained by higher accuracy in the DynArb group, but I still wonder if this would be expected by the model and hold in posterior checks (i.e., if data generated by the best fitting models and parameters for each group subjected to the ME-GLM would recapitulate this pattern of OL and EL weights).

To address this, we have run posterior predictive check analyses, as suggested by the Reviewer (and by Reviewer #3), to try and reproduce the pattern of EL and OL weights from model-generated data. The results aggregated across all participants from the ME-GLM shown in Fig. 2E-F are shown again in Fig. 4A-D together with model predictions. This analysis shows that as a whole (i.e. pooling across all participants within each study), the two arbitration models (fixed mixture and dynamic arbitration) are able to reproduce both EL and OL effects at a magnitude equivalent to the data, whereas the EL model can only reproduce the EL effect and the OL model can only reproduce the OL effect. We have also plotted these model predictions broken down by group in Fig. 6, specifically by using the best-fitting model (and parameters) for each group (i.e. DynArb model for the DynArb group, ExpLearn model for the ExpLearn group, and so on). We show in Fig. 6C-D that we can reproduce the higher weights seen in the data (Fig. 6A-B) for the DynArb group. When we randomly shuffle

the groups (i.e. assign each participant to one of the five groups at random), the difference disappears as expected (Fig 6E-F).

New Figure 4, p.12:

Figure 4. Posterior predictive checks of ME-GLM effects. (A-B) The ME-GLM predicting choice from past outcome (EL effect) and past partner’s action (OL effect) was run on choice data generated with each of the 5 models, using participants’ best-fit parameters. Plotted are the resulting ME-GLM coefficients and their standard error as error bars for Study 1 (A) and for Study 2 (B). Coefficients obtained on the actual data are shown on the left bars for comparison. (C-D) Individual random effects obtained from participants’ data (x-axis) and from model-generated data (y-axis) are shown for Study 1 (C) and for Study 2 (D), together with the best-fit linear regression line and correlation coefficient R, for the EL model (left), OL model (middle) and dynamic arbitration model (right).

Results, p.11: “Posterior predictive checks were also performed on the models using participants’ best-fitting parameters. [...] Through more in-depth simulations, we then proceeded to generate data from each model using participants’ best-fitting parameters, and ran the mixed-effects GLMs shown in Fig. 2E-F (signature of hybrid EL/OL behavior) [...] on the model-generated data. First, examining the effect of past outcome (EL effect) and of past partner’s action (OL effect) on choice (Fig. 4A-B), we found that as expected, the EL effect was well recovered by the EL model and both arbitration models, while the OL effect was well recovered by the OL model and both arbitration models. The baseline model was not able to recover any EL or OL learning effect. Correlations between the data and the model predictions across individuals confirmed this result (Fig. 4C-D), with the EL model accurately predicting the EL but not OL effect, the OL model accurately predicting the OL but not EL effect, and the dynamic arbitration model accurately predicting both effects.”

New Figure 6, p.16:

Figure 6. Group differences in single strategy use and associated model predictions. (A-B) The main effects of past outcome (EL effect, blue) and the effect of past partner’s action (OL effect, orange) on current participant’s choice (previously shown in Fig. 2E-F) are now calculated separately for each group for Study 1 (A) and Study 2 (B). Each dot is an individual participant (random effect); error bars represent SEM. See **Table S4B** for statistics. (C-E) To test whether model-generated data could reproduce this pattern of effects, and to further validate the behavioral signature in the 5 groups, the same ME-GLM as in A-B was run on data generated by each of the 5 models using participants’ best-fitting parameters, then showing the resulting effect for each group using that group’s best-fitting model (C-D) or after assigning participants into the 5 groups at random then using each group’s corresponding model (E-F).

Results p.15: “Additionally, posterior predictive checks confirmed that pattern of GLM effects between groups, whereby GLM effects from model-generated data matched the data well when split by actual groups (Fig. 6C-D), but not when split by randomly shuffled groups (Fig. 6E-F).”

I also wonder whether this could be consistent with the relative model evidence being weaker for ExpLearn and ObsLearn groups compared to the DynArb group. Could the authors provide any additional data to address this, for instance absolute goodness-of-fit metrics (eg, pseudo-R2) and differential model comparison metrics (eg, delta BIC/AIC) for each model/group? What would be the implications if the DynArb group are not only using a unique strategy but are better learners overall (i.e., they differ both qualitatively and quantitatively).

We have now provided AIC and out-of-sample predictive accuracy (which allows assessing absolute goodness-of-fit while partially accounting for overfitting, contrary to pseudo-R2) for each model, averaged across all participants, in **Table 1**. We also show out-of-sample accuracy broken down by model and by group in **Fig. 5A-B**. As suggested by the reviewer, the data does indeed suggest that people in the DynArb group seem to be better learners – this is quite clear when comparing the learning curves between groups (**Fig 5D-E**). The intriguing question from this finding is whether this is completely expected given that DynArb is the most advantageous strategy to solve this task out of the 5 models, hence resulting in better learning, or whether the difference is caused by the individuals that make up the different groups.

After running follow-up analyses, it appears that both may be true. When simulating learning curves from all models (excluding Baseline) and keeping the underlying parameters the same across models (hence assuming the same “simulated” individuals), we still find that DynArb learns faster and makes the best predictions out of all models (**Fig. 5C**). But, when generating data using each participant’s best fitting parameters then plotting the learning curves for each group (**Fig. 5F-G**), the difference between DynArb and the other models appear stronger, suggesting that differences between the individuals that make up each group are responsible. Those individual differences are reflected in the out-of-sample accuracy values in **Fig. 5A-B**, whereby people in the DynArb group (bottom row) exhibit overall higher accuracy values than other groups – in particular, the ExpLearn model performs better in the DynArb group (Study 1=0.618, Study 2=0.590) than in the ExpLearn group (Study 1=0.541, Study 2=0.566), and same is true for ObsLearn (ObsLearn performance in DynArb group: Study 1=0.638, Study 2=0.630; ObsLearn performance in ObsLearn group: Study 1=0.611, Study 2=0.602).

The implication of this is that, somewhat unsurprisingly, for a participant who performs well in this task, DynArb is likely to be the model that will capture that participant’s behavior well. Because ExpLearn, ObsLearn and FixArb are all nested models of DynArb (i.e. by fixing some parameter values, DynArb can default to either of these three models), it is also expected that these other models will perform well in this group.

We now discuss this briefly in the discussion:

Discussion, p.23-24: “The dynamic arbitration model was built to be the most advantageous strategy to solve this task. We confirmed that this was the case using simulated data and posterior predictive checks, showing stronger signatures of learning and of both EL and OL strategy used in data generated by the dynamic arbitration model compared to other models. This explained why participants best fit by the dynamic arbitration models were overall better learners, and reproduced the heterogeneity in behavioral patterns observed across groups.”

New Figure 5, p.14:

Figure 5. Model out-of-sample predictive accuracy and learning curves by group. (A-B) We computed out-of-sample accuracy for each participant across blocks (leaving one block out) and for each of the five models, in Study 1 (A) and Study 2 (B). The top row of the heatmap shows the average predictive accuracy for each model, while the bottom five rows show the breakdown for each group. (C) To ensure that the four learning models (excluding Baseline) were able to learn the task, independent of the participants’ best-fitting parameters, we generated data from those four models using the same simulated parameter values across models and plot the resulting learning curves. The curves reflect the choice accuracy on the first 8 trials following each reversal (similar to Fig. 2A-B). The shaded area represents standard errors across 100 simulations. (D-I) Learning curves were also computed from the actual participants’ data, separately for each group (D-E, see Table S4A for statistics), from model-generated data using each group’s best-fitting model (F-G), and from model-generated data using randomly shuffled group membership (i.e. assigning each participant to a group at random then using this group’s corresponding model; H-I). The shaded area represents standard errors across participants within each group.

2. I was surprised to see the details of the EL model with regard to the incorporation of reward magnitude. Instead of being incorporated into the EV calculation, the magnitude is learned separately and affects choice directly. The authors mentioned that the magnitude effects were unexpected and they seem to have modified the model accordingly, but it is still surprising that this reward magnitude effect is not integrated into the token EV, and is learned separately only biasing choice. I would like to see more justification and model comparison to for this model architecture, as well as discussion on how this may affected the overall model comparison of different strategies.

We thank the reviewer for raising this important point – when initially performing exploratory analyses in Study 1, we had indeed tested different variations of the EL model to assess how magnitude was best incorporated. We had chosen the current EL model to keep in our model set given that it explained the data best of all other EL mechanisms. Given the reviewer’s concern, and also in response to Reviewer 3’s comment about the magnitude boosting mechanism, we have now added this analysis to the Supplementary material, which we hope helps to justify our choice of the current EL model. Note we had run this initial model comparison analysis only on Study 1, because Study 1 was used to set up the model pipeline that we subsequently deployed for Study 2, which we considered as a replication sample.

In particular, on the Study 1 data we tested 3 other EL models:

- ExpLearn_nomag completely ignores reward magnitude and only learns from binary outcomes (essentially coding any reward as a 1 regardless of its magnitude, and any non-reward as a 0, and the model learns the reward probability of each token).
- ExpLearn_mag completely incorporates reward magnitude inside the learning rule (rather than having a separate magnitude boosting mechanism), such that it learns the average reward value (in points) rather than its probability. This model is unlikely to perform well on the task given that magnitude varied from trial to trial in an unpredictable way (therefore could not be learned), but it is still possible that some participants may be trying to learn this way and therefore this model could perform well for those individuals.
- ExpLearn_decay includes the magnitude boosting mechanism as in our main ExpLearn model, but instead of a 50% fixed decay rate applied to unchosen and unrewarded options, this model estimates the decay rate as a fixed parameter.

New Table S3, p.42:

Model	Description	N_{param}	AIC	OOS accuracy
ExpLearn_nomag	Magnitude is ignored, only learning of reward probability and outcomes are treated as binary (1: reward, 0: no reward)	2	212.3	0.524
ExpLearn_mag	Magnitude is incorporated in EV calculation and learned.	2	210.7	0.525
ExpLearn_decay	Magnitude is learned separately and boosts EV, decay rate (free parameter) applied to unchosen and unrewarded tokens.	4	208.7	0.539
ExpLearn	Model variant used in the main analyses reported in the manuscript. It is the same as ExpLearn_decay but with a fixed decay rate of 0.5.	3	207.7	0.539

Table S3. Summary of additional EL models tested during initial exploratory analyses of Study 1 data.

We found that the ExpLearn model performed best when accounting for model complexity, which is why we selected this model as our candidate model for the EL mechanism. We also found that the mean decay rate estimated from the ExpLearn_decay model was not different from 0.5 (mean=0.513, sd=0.227, T(125)=0.639,

$P=0.52$), hence, providing a rationale for why we decided to use a fixed decay of 0.5 in our final EL model, thus minimizing model complexity.

Minor points:

3. Toning down language on psychiatric illness and comorbidity. While the authors are generally cautious about terminology, in a few of places they talk about benefits regarding "comorbid symptom dimensions that may be confounded in the clinic" (line 2, p. 17), "psychiatric profiles" (line 22, p. 19), "symptom dimensions that are typically comorbid in the general population" (line 30, p. 20). I would recommend avoiding the terms comorbidity and psychiatric, because they imply pathological illness. The traits studied here may or may not be pathological. Also, autism spectrum disorder and anxiety disorders are not usually confounded in the clinic in that there is no difficult differential diagnosis that is required. So they are comorbid but to my knowledge there is no actual clinical need to deconfound them. In general, and while I do see the value of this work for computational psychiatry, I think it is more appropriate to use terms like dimensional psychopathology traits/symptoms, and coexisting symptoms/traits, and tone down direct implications of this work for clinical psychiatry (given the need to confirm potential relevance in clinical populations).

Thank you, we have now toned down this language throughout the manuscript. Specifically:

- We renamed Figure 8 (p.20) to *'Group differences in psychiatric symptom dimensions'* (instead of *'psychiatric profiles'*)
- We deleted the part of our section title (p.18) mentioning *'unique, group-specific psychiatric profiles'*.
- We replaced *'help separate comorbid symptom dimensions that may be confounded in the clinic'* at the end of the results section (p.21) by *'help separate symptom dimensions that tend to coexist in the population'*, and similarly in the end of the discussion (p.26), we replaced *'are typically comorbid'* by *'tend to coexist'*.
- In the discussion (p.24), we replaced *'suggesting different psychiatric profiles of the 8 factors described above across groups'* by *'suggesting differences in symptom dimensions between the groups'*.

4. In the discussion, the authors argue for the benefit of including poor learners and that poor performance may be indicative of pathologies of interest. They argue that broad exclusion could lead to a reduction in the range of relevant symptom dimensions associated with the use of non-learner strategies. While in general I agree with their approach and see its benefits, absent from this discussion is the limited interpretability of non-learning strategies. In certain psychopathologies (eg, a severe motivational impairment in schizophrenia or MDD), one may see poor performance and learning that is somewhat generic. It may simply represent poor motivation to engage with the task versus an actual strategy or diminished capacity to perform the task using a more appropriate strategy. In turn, impairments in higher-order cognitive function or memory could lead to worse task construal or rule maintenance, which could end up resulting in a similar phenotype. So poor learning may be a mixed bag with various potential interpretations (and unexplored potential mechanisms), and in itself it may not be informative of well-specified computational failure modes. Therefore, the key advantage of interpretability enabled by theory-driven modeling approaches is in my mind less clearly applicable to this particular group. Perhaps it would be good to provide a more balanced discussion on this point to briefly represent this counterargument.

We thank the reviewer for that suggestion and agree that the interpretability of the non-learning group is more limited than for the other groups in terms of what their behavior might mean and possible mechanistic associations with symptom dimensions. We have now added the following paragraph to the discussion to address this point:

Discussion, p.24: "we do acknowledge that the interpretability of behavior in this group is limited, given that more than one strategy was included in the model and that those strategies aren't necessarily reflective of underlying cognitive mechanisms. Contrary to the other groups whereby associations with symptom dimensions can be directly interpreted in light of the theory-driven modelling approach, in this non-learning group it remains more challenging to characterize what associations might mean, whether they reflect a general deficit in learning, motivational impairments, or reduced working memory capacity, among others."

5. Typo: "different point in times" (line 17, p. 2)

Thank you, we have fixed that.

Reviewer #2:

In this clear and detailed paper, the authors report that individuals use a combination of strategies - experiential and observational learning - in order to maximise rewards and minimise punishments. Importantly, they show that there are subgroups of individuals who use different combinations of these strategies - with some showing their hypothesised 'dynamic arbitration' pattern, in which individuals choose between these strategies depending on the prediction errors received. I think this is a novel paper that addresses an important question, but I have some theoretical and some writing comments.

1. My main theoretical comment is that the two learning strategies actually form different parts of the task - the early part of each trial involves another player making choices, and the individuals having to infer from their choices and the instructions what the other player is intending. This is complex, and relies on many cognitive processes (working memory to recall the instructions and practice to understand what the ***other***** player knows; theory of mind, etc) - other than just experiential learning. Additionally, the player must assume that the other player is learning - another process to account for when making inference about their choices. More comment on this in the discussion might be warranted, as I do not think that the task equates the two 'learning' processes exactly (either in terms of difficulty or confounds).**

While we do argue that the two learning processes are present, we do not claim that they are equivalent in terms of cognitive demands or difficulty. What the reviewer describes about the early part of the trial (making inferences from observing the other player's choice, understanding what the other player knows and assuming that the other player is learning) are exactly the components that make up our observational learning model, so we completely agree with the reviewer that those components constitute something "other than just experiential learning". On the other hand, the experiential learning process relies on learning from the reward (or lack thereof) when the outcome is shown at the end of the trial. Separating the two strategies as different components of the trial is exactly what allows us to quantify and model them separately, even though participants make a single choice on each trial where those components have to be integrated.

That said, we agree with the reviewer that the two processes may not be equivalent in terms of working memory or cognitive demands – interestingly though, despite the reviewer's valid suggestion that OL may be more difficult given the need to infer another player's learning, knowledge and goals, we find overall evidence that OL is favored compared to EL if only considering these two strategies. We have now added a short paragraph in the discussion to comment on this:

Discussion, p.22: "It is possible that the OL and EL strategies aren't matched in terms of complexity, working memory demands, or cognitive processes at play, whereby OL may require additional demands on the learner's part. That said, our finding that across both studies, participants tend more towards using OL compared to EL mitigates this concern about complexity differences. Additionally, our focus on an arbitration mechanism does mean the strategies being arbitrated have to be dissociable and rely on separable computations, in order to best characterize how arbitration is governed. Our task achieved this dissociation, with OL and EL updates taking place at different timepoints in the trial, OL and EL predictions being only weakly correlated, and OL and EL uncertainty trials being separable."

2. Relatedly, here the process of 'dynamic arbitration' is dependent only on the environment, and the environment's uncertainty at that: in many other arbitration processes (e.g. model-based/model-free

learning) arbitrating between strategies relies also on the individual's goals, or other environmental features. This is briefly mentioned in the introduction (end of page 2), and could be expanded on.

We thank the reviewer for the opportunity to expand on that and completely agree that dynamic arbitration processes could be dependent on many other factors. Manipulating uncertainty in this task was done as an initial way to assess and measure the arbitration process in a controlled manner, and it is also consistent with most of the past literature on arbitration and the concept of mixture of experts. That said, and especially in the context of tackling more "social" learning processes, manipulating reliability of the strategies in other, and possibly more naturalistic, ways would be an obvious next step. We have now added a few sentences in the discussion to expand on this:

Discussion, p.23: "it is possible that other factors could drive arbitration. For example, OL reliability may be influenced by observing different partners who vary in performance and expertise, or by manipulating the social context (cooperative versus competitive interaction, incentive for the partner to deceive the participant or to act with a different goal in mind). EL reliability could also vary as a function of stakes being manipulated in a more meaningful way than the current design, or by increasing or decreasing the ability to experientially sample the tokens and their outcomes. Another interesting question would also be to assess how the reliance on each strategy, and arbitration between them, may vary in response to more naturalistic and ecologically valid ways of implementing the strategies in the task. Future work is needed to better characterize the role these factors may play in cross-domain arbitration."

Some more minor points:

*******Abstract*******

3. In the abstract, 'control over behaviour' is assigned to learning strategies. This wording seems somewhat awkward - are these learning processes really also acting as controllers?

We have now changed the wording of that sentence in the abstract to "how do people favor one strategy over the other depending on the environment?". We have also overall shortened the abstract so that it fits under the 200-word limit.

4. I felt that the start of the introduction was framed heavily around social skills, which are important but I don't believe experiential learning only relates to social skills.

We have reworded the start of the introduction so that it better reflects the balance between individual and social learning.

Introduction, p.2: "As humans, we learn about the world around us by seeking and integrating information from multiple sources. On the one hand, we heavily rely on our own past experience to predict the future. Experiential learning (EL) is such that actions that were rewarded in the past tend to be repeated, while actions that were punished in the past tend to be avoided. EL can be relied on to solve many reinforcement learning problems, from learning simple associations between stimulus, action and rewards (model-free learning) to complex cognitive maps (model-based learning) and exploitation/exploration trade-offs¹⁻⁴. On the other hand, as a social species with sophisticated social skills that allows us to make collective decisions and function in society, humans can learn from observing others⁵⁻⁷. Such observational learning (OL) is thought to confer the evolutionary advantageous ability to assess the consequences of actions available in the environment without

having to directly experience the potentially negative outcome of those actions. OL is prevalent across many domains, from basic sensory-motor learning⁸⁻¹⁰ to complex strategic decision-making^{11,12}, and from aversive^{13,14} to reward learning¹⁵⁻¹⁸.”

5. Typo: Should be explosion *of*** online data collection, I think (line 6, page 3)**

Thank you, we have fixed that typo.

6. The link between online testing and different strategies seemed a little forced - surely people use different strategies in person too? Maybe this point could be made clearer?

We completely agree with the reviewer. The advantage of online studies for examining heterogeneity is that they can more easily provide appropriately powered large sample sizes, and the opportunity for independent replication, compared to in-person studies, which are often smaller and sometimes made of more homogenous (e.g. convenience) samples. But we completely agree that the heterogeneity would also exist in person. We have now made that clearer:

Introduction, p.3: “Though such heterogeneity is likely to exist in any study sample (online and in-person studies, clinical and general populations, etc), it is not usually well characterized in existing studies, given that sample sizes are too small or that most computational modelling approaches tend to select a “winning” model and apply it to all participants.”

Results

7. Uncertainty is defined in quite a basic way, based on the last few trials. Of course modelling is used eventually, but even in this case uncertainty is defined based on prediction errors. Maybe this could be commented on in the discussion - more sophisticated estimates of uncertainty, based on more than just a couple of trials back, could have been used.

We note that based on Reviewer 3’s comments, we now also provide evidence that our behavioral results hold when using the uncertainty conditions as defined by the task design (see new **Figure S1**), though those block conditions are not as strongly predictive of choice as the trial-by-trial definition we report (see new **Table S2**). This helps provide further justification for using our current definition of uncertainty, which reflects the fact that things are changing quickly in the task and participants are likely to experience uncertainty varying from trial to trial.

We do agree, however, that other, more sophisticated definitions of uncertainty, or modelling of reliability, could also be tested and we have now added this point to the discussion:

Discussion, p.22-23: “As part of our study design, we decided to focus on uncertainty as the main factor driving the reliability of each strategy. [...] we identified that our specific trial-by-trial definitions of uncertainty were better predictors of choice than the uncertainty blocks. That said, our definition was based on the immediately preceding sequence of trials and could have involved a more sophisticated definition of uncertainty over longer timescales.”

8. Some figures seem to have boxes around them, including in the supplement, which should be fixed if possible.

Thank you, we have now saved higher quality versions of the figures and ensured the boxes that appeared before are now gone.

9. In the ME-GLM, I wasn't sure whether significant effects could be seen from trials where EL and OL would produce a consistent result - i.e., are these effects really specific to EL and OL? In the first analyses only trials that are consistent with one and not the other are used, presumably to avoid this issue.

The reason why we can quantify the unique contribution of EL and OL effects on behavior in the ME-GLM is because the EL and OL regressors (i.e. effect of past outcome and effect of past partner's action) are included together, and therefore compete for variance in explaining choice. The reviewer is right that because we include all trials in the ME-GLM, there is some positive correlation between the EL and OL regressors (to be exact: $R=0.262$ in Study 1, $R=0.272$ in Study 2, now reported in Methods, p.29). That said, those numbers indicate that the shared variance is low (i.e. around 7-8%), meaning we can be quite confident that the resulting EL and OL effects are specific and can be reliably dissociated in this analysis.

10. I disagree with the use of 'marginal' to describe non-significant p-values - these should not be reported (pages 15-16).

We have now removed all mentions of "marginal" effects throughout the manuscript.

11. Typo: I think it should say 'trials' not 'trial' (line 2, page 24)

Thank you, this has been corrected.

12. Could the authors comment on whether there is enough power to draw strong conclusions about the differences between groups? In particular, some groups only have a very small number of individuals within them.

To address this concern, we have conducted post-hoc power analyses for Study 1 specifically (since the sample size is much smaller than in Study 2), for all results where we report a difference between groups. For all analyses we used $\alpha=0.05$, a total sample size of 126, and 5 groups:

- Figure 5D (group differences in learning curves): $F(4,875)=13.49$, $\eta_p^2=0.058$, resulting in achieved power for 8 measurements (time points) > 0.99
- Figure 6A (group differences in OL and EL ME-GLM effects): $F(4,125)=25.55$, $\eta_p^2=0.450$, resulting in achieved power for 2 measurements (OL vs EL effects) > 0.99
- Figure 7A (group difference in arbitration index): $F(4,112)=17.13$, $\eta_p^2=0.380$, resulting in achieved power (one-way ANOVA) > 0.99
- Figure 7C (group differences in OL and EL ME-GLM effects for high vs low OL uncertainty): $F(4,375)=20.22$, $\eta_p^2=0.177$, resulting in achieved power for 4 measurements (2-by-2 strategy*OL uncertainty) > 0.99

- Figure 7D (group differences in OL and EL ME-GLM effects for high vs low EL uncertainty): $F(4,375)=19.90$, $\eta_p^2=0.175$, resulting in achieved power for 4 measurements (2-by-2 strategy*EL uncertainty) > 0.99

For Figure 8A-E (group differences in psychiatric symptom dimensions), we used a total sample size of 568 (pooled data after careless exclusion): $F(28,3948)=2.38$, $\eta_p^2=0.017$, resulting in achieved power for 5 groups and 8 measurements (factors of symptom dimensions) > 0.99.

Overall, these analyses show appropriate power for the effect sizes reported.

Discussion

13. If reward magnitude effects are only in the supplement, they shouldn't be discussed in main text - either this discussion should be in the supplement, or the results of the reward magnitude analyses should be presented in the main text.

We have now moved all the discussion about reward magnitude effects to the Supplement (p.48).

14. I agreed with the points made re: baseline models and including data/relevance for psychopathology.

Thank you for this! While the interpretability of baseline models is limited in terms of underlying computational mechanisms (see Reviewer 1's comment), we hope this encourages other researchers to at least consider those models and include them in their findings.

Methods

15. Was the exclusion criteria (25% of trials) pre-planned? It also isn't clear from the methods that participants were also excluded for missing catch questions - I would perhaps appreciate a sensitivity analysis to check the exclusion of these participants aren't affecting these results.

We decided on the exclusion criterion of 25% of missed trials based on a previous study using an observational learning task (Charpentier et al, 2020, *Neuron*), where we used 20% of missed trials, and made this threshold slightly more lenient (25%) given our expectation that online study behavioral data would be noisier, all the while ensuring that enough data was obtained for each participant to be able to fit the computational models. We also ensured that this criterion made sense after Study 1 data collection, and kept it the same for Study 2.

As for missing catch questions, this goes into determining exclusion only for the psychopathology analyses. Indeed, catch questions were only included in some of the questionnaires, and their presence wasn't necessarily associated with performance on the task (since a participant may perform well during the task and stop paying attention at the end during the questionnaire – we therefore deemed that it would be too strict a criterion to exclude participants' task data just based on missed catch questions during the questionnaires). The procedure for excluding participants based on questionnaire data (missed catch questions as well as careless responding) is detailed in the Methods (*Careless exclusion* paragraph, p.35).

There were 16 participants excluded in Study 1 and 35 in Study 2 (based solely on responding during the questionnaires). Performance on the task did not differ between these participants and those who didn't miss any catch question - see the two tables below for completeness.

Study 1 (N=126, N_{included} = 110, N_{excluded} = 16, df for t-test is 124):

	Included subjects (N=110)		Excluded subjects (N=16)		Tdiff	Pdiff
	Mean	SD	Mean	SD		
Accuracy	0.582822	0.083226	0.578331	0.109866	0.193204	0.847115
RT	618.5434	162.6085	662.3654	173.1279	-0.99917	0.319658
Learning slope	0.010189	0.024431	0.008882	0.024336	0.200098	0.841732
Prop_OL_ch	0.516271	0.092367	0.504657	0.113065	0.456363	0.648927
EL_eff_GLM	0.357864	0.519851	0.280542	0.413231	0.568709	0.570582
OL_eff_GLM	0.217831	0.200753	0.180944	0.200689	0.686746	0.493525
arbitration_index	0.34441	0.442201	0.337208	0.400633	0.061444	0.951107

Study 2 (N=493, N_{included} = 458, N_{excluded} = 35, df for t-test is 491):

	Included subjects (N=458)		Excluded subjects (N=35)		Tdiff	Pdiff
	Mean	SD	Mean	SD		
Accuracy	0.60446	0.087223	0.594339	0.10244	0.653079	0.514011
RT	611.7215	146.4916	607.9697	210.2509	0.140959	0.88796
Learning slope	0.008487	0.02384	0.012831	0.021522	-1.04565	0.296236
Prop_OL_ch	0.490569	0.107165	0.519755	0.094874	-1.56475	0.118287
EL_eff_GLM	0.511574	0.47449	0.463063	0.553301	0.57586	0.564974
OL_eff_GLM	0.238151	0.167534	0.233198	0.191733	0.1668	0.867596
arbitration_index	0.392362	0.416032	0.428838	0.459805	-0.48901	0.625057

We now report the following in the manuscript:

Methods, p.35: “Note that performance on the task was not different between these later excluded participants and all remaining participants, hence why they were only excluded at this stage of the analysis.”

16. What does deemed exempt mean with regards to ethical approval?

This just means that the study was deemed exempt from a full in-depth IRB review following an initial review conducted by Caltech’s IRB committee. This “exempt” status is allowed under US Federal IRB regulations if the proposed study is deemed no more than minimal risk and several other criteria are met. In our case, these include that we are only recruiting adults from the general population, and that we are not collecting any identifiable information from participants (because the study was run on Prolific and we did not access participants’ names, contact information, exact date of birth, etc). When a study is deemed exempt, it does not require the same level of in-depth evaluation and monitoring as a study subject to full IRB review, hence minimizing administrative overhead. To avoid any confusion, we have added the following clarification:

Methods, p.26: “All participants provided informed consent. The study was deemed exempt from full IRB review after an initial review by the Caltech Institutional Review Board due to it being judged to be of minimal risk to participants and meeting several other criteria required for an exempt status.”

17. The choice of 50% decay based on ‘previous analyses’ isn’t well-justified - perhaps this could be expanded on in the Supplement.

We have now added to the supplement those initial exploratory analyses where we tested different versions of the EL model on Study 1 data (exploratory analyses before replication in Study 2) to better justify how we chose the current model with a decay rate of 0.5, as also suggested by Reviewers #1 and #3.

New Table S3, p.42:

Model	Description	N_{param}	AIC	OOS accuracy
ExpLearn_nomag	Magnitude is ignored, only learning of reward probability and outcomes are treated as binary (1: reward, 0: no reward)	2	212.3	0.524
ExpLearn_mag	Magnitude is incorporated in EV calculation and learned.	2	210.7	0.525
ExpLearn_decay	Magnitude is learned separately and boosts EV, decay rate (free parameter) applied to unchosen and unrewarded tokens.	4	208.7	0.539
ExpLearn	Model variant used in the main analyses reported in the manuscript. It is the same as ExpLearn_decay but with a fixed decay rate of 0.5.	3	207.7	0.539

Table S3. Summary of additional EL models tested during initial exploratory analyses of Study 1 data.

We found that the ExpLearn model performed best when accounting for model complexity, which is why we selected this model as our candidate model for the EL mechanism. We also found that the mean decay rate estimated from the ExpLearn_decay model was not different from 0.5 (mean=0.513, sd=0.227, $T(125)=0.639$, $P=0.52$), hence, providing a rationale for why we decided to use a fixed decay of 0.5 in our final EL model, thus minimizing model complexity.

18. The data and code statement is not up to date in the main text - this is already available I think.

Thank you for pointing this out. We have now updated that statement to reflect that the data and code are already available online at https://github.com/ccharpen/OL_EL_behavior.

Supplement

19. For parameter recovery, it would be clearer if white was 0, and red/blue were the opposite ends of the spectrum - strong negative correlations between parameters should not appear white!

Thank you for the suggestion – we now use a blue to red scale, with white showing $R=0$ correlations (**Fig. S3**).

20. Figure S6a looks like 7 factors might have a lower BIC? Perhaps presenting the BIC numbers alongside would be helpful here.

We have added a sentence in the Figure legend to clarify the values. Note that because of additional figures, the figure is now **Figure S8A**.

Figure S8 legend, p.54: "Specifically, BIC for 8 factors was -59393, while BIC for 7 and 9 factors (the next two closest values) was -59379 and -59172, respectively."

Reviewer #3:

In this paper, Charpentier and colleagues investigate the arbitration between experiential and observational learning in 2 samples of human participants. They designed a task in which participants have to select one of 2 colored tokens. One of the colored tokens is (probabilistically) more rewarded, and this changes over time (actually once per block of 20 trial). Participants can figure out how to optimize their behavior from 1) their past history of choices and rewards (experiential), and 2) observing another participant selecting boxes probabilistically associated with token (observational). Because this other participant knows which token is most rewarded, the combination of their choices and past outcomes can be informative about the identity of the best token.

Overall, the topic is interesting, and the paper features some high-quality characteristics, such as the large number of participants and the presence of a replication sample.

I nonetheless have serious concern about the framing of the study, the modelling and the robustness of the conclusions, which make me doubt the appropriateness of the current version of this manuscript for Nature Communications. I hope my comments will be helpful to the authors to improve this manuscript, which builds on a potentially interesting setup.

We thank the reviewer for noting the positive features of the study and for providing constructive feedback to appropriately revise the manuscript and address the concerns. We believe the manuscript has been substantially improved as a result of this feedback and suggestions.

1. First and most critically, in some instances, the analytical strategies which support the current story do not seem to fit with the actual experimental design, making me quite uncomfortable with the whole piece. From what I understand, the actual experimental design is a repetition of blocks of 20 trials (each with new pairs of fractals) featuring a fixed OL uncertainty level and one reversal in the EL contingencies. It is definitely not a symmetrical manipulation of OL and EL levels of uncertainty (changes in levels of OL uncertainty are clearly announced by a change of cues; changes in levels of EL uncertainty have to be detected by trials-and-errors), contrary to what the manuscript seems to suggest with the “arbitration” story. Similarly, changes in OL uncertainty cannot be conceptualized as “switches”, because each pair of fractals have a fixed box-to-token transition probability (though it is not clear if, e.g. Fig 4 & C consider as “switches” only the proper reversal in EL, or all changes in EL and OL contingencies). In the end, for most behavioral analyses, the authors rely on a very arbitrarily defined notion of uncertainty for EL and OL [p5: EL uncertainty (blue points) was deemed low on trials where past outcome-action-outcome sequence was consistent, and high otherwise. OL uncertainty (orange points) was deemed low if the past two box-token transitions were consistent, and high otherwise], and the writing seem to suggest that those have been experimentally manipulated over the course of the experiments. I would certainly have more faith in a set of analysis that explore the role of properly manipulated factors (i.e. actual levels of probability) rather than of those arbitrary metrics. Those concern may seem trivial, but I think they actually jeopardize a significant share of the narrative that the proposed analyses are meant to support.

We agree with the reviewer that the underlying manipulation of contingencies is not completely symmetrical between the two strategies. We designed the task in that way because of the possibility that OL requires enhanced cognitive demands (learning of box-to-token transition probability followed by inference about the goal token from observing the partner's chosen box) compared to EL (learning of token to reward associations only). Therefore, this design choice to incorporate longer OL blocks would allow for potential slower learning in

OL than EL. In practice, we do not see that pattern in the data, so future versions of the task could indeed better reduce this asymmetry. Additionally, while the reviewer is right that changes in fractals provide an explicit cue that resets learning of the box-to-token transition probabilities, we would like to emphasize that the changes in uncertainty conditions (for both OL and EL strategies) were not cued. For OL, the change in fractals implies a reset in learning, as mentioned above, but it does not cue a change in the OL uncertainty level, in the sense that participants do not automatically know whether OL uncertainty has changed when they see a new pair of fractals. Similarly, for EL, while there is no reset of learning in the same way as the change in fractals (the two tokens visually remain the same throughout the task), participants do know that reversals can happen given the experiential practice trials that they complete during the instructions, and changes in EL uncertainty levels (80/20 vs 40/60 contingencies) remain implicit and uncued. So while our design (by choice) contained some asymmetry between the two strategies to account for potential complexity differences, changes in both OL and EL uncertainty levels were never explicit, hence why we believe that using the pre-defined uncertainty conditions would not necessarily reflect perceived uncertainty from the participant's perspective. We now acknowledge this potential limitation in the discussion (see below) and hope to address the reviewer's concern with the following points.

First, to clarify the terminology about the use of "switches" in the text, we now exclusively refer to "reversals" to characterize the changes in token reward probability that led to the other token becoming more valuable (so for example when blue was more valuable, whether at 60 or 80% reward probability, then orange becomes more valuable, again regardless of uncertainty level). This is what "switch" meant in Figure 2A-B and Figure 4A-B (now **Figure 5C-I**), which we have now changed to "reversal". Therefore, "reversals" refer to changes in the most valuable token, while we now exclusively use "change in EL uncertainty condition" to refer to the 80/20 (low uncertainty) versus 60/40 (high uncertainty) periods, which by design are orthogonal to the identity (blue or orange) of the most valuable token.

Second, we apologize for the confusion whereby the mention of EL and OL uncertainty during the results section refers to specific trials extracted in the task to reflect specific situations where we expect each strategy's predictions to be more or less reliable. To address that, we now refer to "EL/OL uncertainty **conditions**" when describing the manipulated factors in the design and "EL/OL uncertainty **trials**" when describing the specific trials in the task where participants' experienced level of uncertainty is likely to vary the most, regardless of the conditions we established in the design.

Third, the reviewer's most significant concern here seems to be the motivation for relying on EL/OL uncertainty **trials** instead of **conditions** during the analysis. The reason why we focused on trials as opposed to the conditions *per se*, is that while the conditions were specifically introduced to perturb uncertainty as a key experimental manipulation (and as we now show these manipulations indeed worked as planned – see below), from the perspective of an individual participant, the participant does not know when uncertainty has changed according to the experiment (because the changes in uncertainty are not signaled). Instead, they must **infer** changes in uncertainty based on their own experience of the task. Variation in experienced probabilities across trials will thus naturally mean that there is no simple boundary between a participant's subjective experiences of high and low uncertainty for each of the experimental variables, and that this will not necessarily match with the programmed boundary changes associated with our experimental condition manipulations.

We now provide more motivation for this in the text (see below), to further justify the rationale behind why and how we defined those uncertainty trials. Following up on this rationale, we ran several analyses to confirm that defining specific uncertainty trials was more representative of the actual uncertainty that participants experienced during the task, and that this impacted their behavior more strongly than the uncertainty conditions from the task design. Those analyses yield the following results:

- First, as expected, the two definitions of uncertainty were associated, such that there were more low uncertainty trials in the low (than high) uncertainty condition, and vice versa (difference in proportion of OL uncertainty trials: Study 1: $T(125)=30.3$, Study 2: $T(492)=61.1$; difference in proportion of EL uncertainty trials: Study 1: $T(125)=52.2$, Study 2: $T(492)=91.5$; all $P_s < 0.0001$).
- Furthermore, we ran two ME-GLMs to test whether choice was better predicted by low vs high uncertainty **trials** or by low vs high uncertainty **condition** (i.e. uncertainty definition by design). In the first GLM, we predicted OL choice (i.e. whether choice on each trial is consistent with OL or not – regardless of EL) from OL uncertainty (low vs high) as defined by our trial-by-trial definition, or by the design block conditions. In the second GLM, we repeated the above to predict EL choice from EL uncertainty trials and conditions. For both GLMs, and in both studies, we found that the trial definition of uncertainty was a stronger predictor of choice than the condition definition from the task design, thus further justifying its validity. We now present this justification in **Table S2**.
- Finally, following the reviewer’s suggestion, we have re-run all behavioral analyses using the uncertainty conditions (rather than uncertainty trials) and report them in the Supplemental materials. Specifically, we have now added **Figure S1**, which shows the same analysis as in Figure 3, the only difference being the definition of uncertainty as the blocks in the design (uncertainty **conditions**) rather than uncertainty **trials**. We still find all the effects of uncertainty conditions in the expected direction, simply weaker than when using uncertainty defined on a trial-by-trial basis. These weaker effects may be accounted for by a lag in the information integration given that changes in uncertainty were uncued (potentially leading to participants assuming they are still in a low uncertainty block for a while after transitioning to a high uncertainty block or vice-versa), as well as because high uncertainty trials are still present in low uncertainty block and vice-versa.

*Results, p.7: “First, we classified trials as low versus high OL uncertainty trials and low versus high EL uncertainty trials depending on the recent trial history (**Fig. 1B**, see **Methods** for details). Those trials broadly overlapped with the low vs high uncertainty conditions that were defined by design (larger proportion of low OL uncertainty trials in low compared to high OL uncertainty conditions: Study 1: $T(125)=30.3$, Study 2: $T(492)=61.1$; larger proportion of low EL uncertainty trials in low compared to high EL uncertainty conditions: Study 1: $T(125)=52.2$, Study 2: $T(492)=91.5$; all $P_s < 0.0001$), but were defined to capture trial-by-trial variations in uncertainty. We hypothesized those variations would be more representative of how dynamic changes in uncertainty were experienced by participants, given that actual changes in uncertainty were not cued, which would lead to a lag in information integration when considering the blocked conditions. Indeed, we found that uncertainty trials were stronger predictors of choice throughout the task than uncertainty conditions (**Table. S2**).”*

Methods, p.27: “This design resulted in OL uncertainty blocks that lasted longer than EL uncertainty blocks, which was done to account for the possibility that OL would be slower than EL (given increased cognitive demands on the social inference process required in OL).”

Discussion, p.22-23: “As part of our study design, we decided to focus on uncertainty as the main factor driving the reliability of each strategy. We manipulated uncertainty across different blocks, with EL uncertainty levels varying within each OL uncertainty block to allow for the possibility of slower learning during OL. The data did not confirm this (if anything learning was faster during OL than EL), suggesting that future designs could better ensure uncertainty periods are symmetrical across strategies. That said, the uncertainty conditions were not used in the analyses since we identified that our specific trial-by-trial definitions of uncertainty were better predictors of choice than the uncertainty conditions, thus mitigating the design asymmetry concern.”

New Table S2, p.42:

A) OL choice $\sim 1 + OLunc_{trial} + OLunc_{condition} + (1 + OLunc_{trial} + OLunc_{condition} | subID)$

Predictor	Study 1			Study 2		
	Estimate \pm SE	t(19991)	95% CI	Estimate \pm SE	t(77960)	95% CI
Intercept	-0.116 \pm 0.042	-2.77	[-0.20, -0.03]	-0.093 \pm 0.021	-4.50	[-0.13, -0.05]
OL unc trial	0.726 \pm 0.090	8.04	[0.55, 0.90]	0.780 \pm 0.041	19.07	[0.70, 0.86]
OL unc condition	0.109 \pm 0.035	3.11	[0.04, 0.18]	0.150 \pm 0.019	8.12	[0.11, 0.19]

B) EL choice $\sim 1 + ELunc_{trial} + ELunc_{condition} + (1 + ELunc_{trial} + ELunc_{condition} | subID)$

Predictor	Study 1			Study 2		
	Estimate \pm SE	t(19991)	95% CI	Estimate \pm SE	t(77960)	95% CI
Intercept	0.002 \pm 0.026	0.08	[-0.05, 0.05]	0.094 \pm 0.014	6.53	[0.07, 0.12]
EL unc trial	0.391 \pm 0.045	8.42	[0.30, 0.48]	0.465 \pm 0.025	18.41	[0.42, 0.51]
EL unc condition	0.025 \pm 0.033	0.76	[-0.04, 0.09]	0.058 \pm 0.016	3.61	[0.03, 0.09]

Table S2. Summary of ME-GLMs predicting choice from uncertainty trials and uncertainty conditions.

New Figure S1, p.46:

Figure S1. Behavioral signature of uncertainty-driven arbitration between EL and OL. This figure is meant to be compared with main text **Figure 3**, and shows the same analysis except that it uses the condition (or block) definition of OL and EL uncertainty – that is, the manipulation of contingencies established in the task design – instead of the trial-by-trial definition of uncertainty, meant to be more representative of changes of uncertainty as directly experienced by the participants. As a

sanity check, this analysis shows that the OL and EL uncertainty conditions do influence choice in the expected direction. However, the effects are weaker than when examining trial-by-trial uncertainty variations (see **Figure 3** for comparison).

2. Second, though apparently sophisticated, I found the whole modelling section uncharacteristically weakly supported by the data. At no point do the author show model fits (i.e. model predictions/fits against actual data, e.g. learning curves), replications of behavioral effects/analyses with model fits (e.g. the GLME featured in Fig 2-3, but with model simulations), or interpretable indexes of model goodness of fit (or absence thereof). Only Figure S3 provides some results on posterior predictive checks but on such an aggregated level that it is hard to feel confident that the model even remotely fit and reproduce participants behavioral data when it comes to the effects of experimental manipulations. I would strongly suggest that the authors 1) show model fits supported by data when appropriate (e.g. learning curves, etc.) 2) provide model simulations that show that synthetic data can replicate all the effects of the experimentally manipulated factors on behavior, as shown on the GLME on behavior (effects of high vs low EL vs OL uncertainty + magnitude), 3) provide estimate of goodness of fit, such as average (out-of-sample) likelihood per trial (of note, these model quality-check analyses seem to have extensively been performed in a previous manuscript of the same team (Charpentier, et al 2020 Neuron)).

We have now performed several analyses to address the reviewers' three points, which we agree are critical in justifying the robustness of our modeling approach:

1) We now show model predictions of learning curves and our main effect GLME (i.e. main OL effect and main EL effect, regardless of uncertainty), shown next to the data and confirming that the models can reproduce the data well. Learning curves are shown in **Figure 5** (data: **Fig. 5D-E**, model predictions: **Fig. 5F-G**), and main GLME effects are shown in **Figure 4A-D**.

2) We have also performed these same analyses of the GLMEs capturing the effects of OL and EL uncertainty trials in **Figure 4E-F**, and correlations between actual and model-predicted effects in **Figure S5** (for both uncertainty **trials** definition and uncertainty **conditions** definition).

3) Finally, we provide additional measures of goodness-of-fit (AIC, out-of-sample accuracy) for each model, aggregated for all participants in **Table 1**, and out-of-sample accuracy broken down by group in **Figure 5A-B**, as also suggested by Reviewer #1.

We copy the new Figures 4, 5, and S5, Table 1, as well as added text below for the reviewer's reference.

*Methods, p.33-34: "we generated choice data for all five models (also from each participant's best-fit parameters) and ran the ME-GLMs (shown in **Fig. 2E-F**) that estimate the main effects of past outcome (EL effect) and past partner's action (OL effect) separately on each model's generated choices. We report the main effects for the data next to the model predictions (**Fig. 4A-B**) as well as correlations between the data and model prediction across individuals for the EL, OL and dynamic arbitration model (**Fig. 4C-D**). Third, to ensure that the dynamic arbitration model could appropriately capture the effect of uncertainty on each strategy, we ran the ME-GLMs (shown in **Fig. 3C-D**) that estimate the EL and OL effects separately for low and high EL and OL trial uncertainty on data generated by the dynamic arbitration model and show the resulting ME-GLMs individual estimates (random effects) next to those estimated from the data (**Fig. 4E-H**). The correlations between ME-GLM random effect estimates from participants data and from model-generated data for this analysis were also computed, separately for each effect and each uncertainty type (**Fig. S5**)."*

New Figure 4, p.12:

Figure 4. Posterior predictive checks of ME-GLM effects. A-B. The ME-GLM predicting choice from past outcome (EL effect) and past partner’s action (OL effect) was run on choice data generated with each of the 5 models, using participants’ best-fit parameters. Plotted are the resulting ME-GLM coefficients and their standard error as error bars for Study 1 (A) and for Study 2 (B). Coefficients obtained on the actual data are shown on the left bars for comparison. C-D. Individual random effects obtained from participants’ data (x-axis) and from model-generated data (y-axis) are shown for Study 1 (C) and for Study 2 (D), together with the best-fit linear regression line and correlation coefficient R, for the EL model (left), OL model (middle) and dynamic arbitration model (right). E-H. To ensure the dynamic arbitration model can reproduce the effect of uncertainty on behavior, we ran the ME-GLM predicting choice from past outcome (EL effect) and past partner’s action (OL effect) on choice generated by the dynamic arbitration model separately for low and high OL uncertainty (Study 1: E, Study 2: G) and for low and high EL uncertainty (Study 1: F, Study 2: H). The lines depict the effect of low vs high uncertainty, error

bars are the standard error of the ME-GLM coefficients, and dots are individual random effects. Effects from the actual data are shown on the left panel for comparison.

New Figure 5, p.14:

Figure 5. Model out-of-sample predictive accuracy and learning curves by group. A-B. We computed out-of-sample accuracy for each participant across blocks (leaving one block out) and for each of the five models, in Study 1 (A) and Study 2 (B). The top row of the heatmap shows the average predictive accuracy for each model, while the bottom five rows show the breakdown for each group. C. To ensure that the four learning models (excluding Baseline) were able to learn the task, independent of the participants' best-fitting parameters, we generated data from those four models using the simulated parameter values across models and plot the resulting learning curves. The curves reflect the choice accuracy on the first 8 trials following each reversal (similar to Fig. 2A-B). The shaded area represents standard errors across 100 simulations. D-I. Learning curves were also computed from the actual participants' data, separately for each group (D-E, see Table S4A for statistics), from model-generated data using each group's best-fitting model (F-G), and from

model-generated data using randomly shuffled group membership (i.e. assigning each participant to a group at random then using this group's corresponding model; H-I). The shaded area represents standard errors across participants within each group.

New Figure S5, p.51:

Figure S5. Correlation between actual and model-generated change in ME-GLM effects with uncertainty. The ME-GLMs computing the interactions between each strategy (EL effect, OL effect) and each uncertainty type (EL uncertainty, OL uncertainty) were run on data generated by the dynamic arbitration model as well as on the actual data (see main text Fig.

4E-H for effects). We then computed, for each participant, the difference in EL effect (blue, **A-B & E-F**) and the difference in OL effect (orange, **C-D & G-H**) between low and high OL uncertainty (**left panels**) and between low and high EL uncertainty (**right panels**). We repeated this analysis using the trial definitions of uncertainty (**A-D**) and the conditions defined by the task design (**E-H**). We found that in both cases, the dynamic arbitration model was able to recover the effects from the data, with stronger recovery for the effect of uncertainty on its corresponding strategy (i.e. EL uncertainty on EL effect – **blue, right panels** – and OL uncertainty on OL effect – **orange, left panels**) and stronger recovery when using uncertainty trials compared to uncertainty conditions to define uncertainty.

Table 1, p.10:

Model	Study 1					Study 2			
	N _{par}	AIC	OOS acc	Frequency	N _{best} (% _{tot})	AIC	OOS acc	Frequency	N _{best} (% _{tot})
Baseline	4	215.7	0.521	0.205	25 (19.8)	219.8	0.511	0.159	83 (16.8)
Experiential learning	3	207.7	0.539	0.147	21 (16.7)	204.9	0.552	0.060	24 (4.9)
Observational learning	2	197.2	0.569	0.311	40 (31.8)	194.8	0.575	0.190	95 (19.3)
Fixed mixture	6	191.0	0.593	0.115	14 (11.1)	186.0	0.607	0.324	160 (32.5)
Dynamic arbitration	6	191.2	0.595	0.222	26 (20.6)	187.1	0.608	0.267	131 (26.6)

Table 1. Summary of model fits. Each of the five models (N_{par} = number of parameters) was fitted to participants data first using Matlab's *cbm* toolbox. Using individual model-fitting, we computed the mean AIC as well as mean out-of-sample accuracy (OOS acc) across participants. OOS accuracy was calculated for each individual by fitting the model on 7 task blocks and using the best-fitting parameters to calculate the likelihood of predicting the participant's choices in the remaining block (then iterating across all 8 blocks). We then used *cbm*'s hierarchical Bayesian inference fitting across all five models to compute model frequency. Selecting the best-fitting model for each individual participant (highest model responsibility), we then calculated the number and proportion of participants for whom each model explains their data best (N_{best} column).

Results, p.11: "Through more in-depth simulations, we then proceeded to generate data from each model using participants' best-fitting parameters, and ran the mixed-effects GLMs shown in **Fig. 2E-F** (signature of hybrid EL/OL behavior) and in **Fig. 3C-D** (effect of EL and OL uncertainty trial types) on the model-generated data. First, examining the effect of past outcome (EL effect) and of past partner's action (OL effect) on choice (**Fig. 4A-B**), we found that as expected, the EL effect was well recovered by the EL model and both arbitration models, while the OL effect was well recovered by the OL model and both arbitration models. The baseline model was not able to recover any EL or OL learning effect. Correlations between the data and the model predictions across individuals confirmed this result (**Fig. 4C-D**), with the EL model accurately predicting the EL but not OL effect, the OL model accurately predicting the OL but not EL effect, and the dynamic arbitration model accurately predicting both effects. Second, we predicted that the uncertainty effects, i.e. the extent to which each strategy use varies with EL and OL uncertainty trials, should be appropriately recovered by the dynamic arbitration model, since this is the only model that explicitly modulate strategy weights based on uncertainty. And indeed, we found that the interactions between strategy use and uncertainty in data generated by the dynamic arbitration model (**Fig. 4E-H right**) matched those observed in the data (**Fig. 4E-H left**), with the model showing a clear effect of OL uncertainty on the OL effect (**Fig. 4E & 4G**) and of EL uncertainty on the EL effect (**Fig. 4F & 4H**). Correlations between the data and model predictions across individuals also showed strong recovery for the effect of

uncertainty on the corresponding strategy (change in EL effect for low vs high EL uncertainty trials – data vs model predictions: Study 1: $R=0.795$, Study 2: $R=0.870$; change in OL effect for low vs high OL uncertainty trials – data vs model predictions: Study 1: $R=0.867$, Study 2: $R=0.886$; **Fig. S5A-D**). Interestingly, we also found that when running that same posterior predictive check analysis with the condition definition of OL and EL uncertainty (instead of trial-by-trial definition), the predictions of the dynamic arbitration model were not as strongly correlated with the data (change in EL effect for low vs high EL uncertainty condition – data vs model predictions: Study 1: $R=0.588$, Study 2: $R=0.712$; change in OL effect for low vs high OL uncertainty condition – data vs model predictions: Study 1: $R=0.633$, Study 2: $R=0.593$; **Fig. S5E-H**). This further validates the uncertainty trial definitions shown in Fig. 1B.”

Results, p.13: “we found that averaging the five models’ out-of-sample accuracy separately for each group confirmed that each group was best fit by its respective model (**Fig. 5A-B**).”

Results, p.13: “Learning curves generated from the four learning models with completely hypothetical parameters (i.e. not the best-fitting parameters) confirmed that learning occurs in those models, and that the dynamic arbitration model produced the fastest learning (**Fig. 5C**). Examining learning curves generated from each group’s best-fitting model (using individual participant’s best-fitting parameters within that group) showed an almost perfect match to the data (**Fig. 5F-G** compared to **Fig. 5D-E**, respectively), which was not observed when group membership was randomly shuffled (**Fig. 5H-I**).”

3.a. Other modelling various modelling issues. Like the behavioral analyses, the model design seems to overly rely one somewhat arbitrary decisions (like the magnitude mechanism, or the computation of EL and OL reliability). At the very minimum, I would like to see some analyses that the chosen implementations actually succeed in capturing the said mechanisms/effects.

As also suggested by Reviewer #1, we now report in the supplemental materials our initial analyses that have led to the choice of the magnitude mechanism. When initially performing exploratory analyses in Study 1, we had tested different variations of the EL models to assess how magnitude was best incorporated. We had chosen the current EL model to keep in our model set given that it explained the data best of all other EL mechanisms. Note we had run this initial model comparison analysis only on Study 1, because Study 1 was used to set up the model pipeline that we subsequently deployed for Study 2, which we considered as a replication sample. In particular, we tested 3 other EL models:

- ExpLearn_nomag completely ignores reward magnitude and only learns from binary outcomes (essentially coding any reward as a 1 regardless of its magnitude, and any non-reward as a 0, and the model learns the reward probability of each token).
- ExpLearn_mag completely incorporates reward magnitude inside the learning rule (rather than having a separate magnitude boosting mechanism), such that it learns the average reward value (in points) rather than its probability. This model is unlikely to perform well on the task given that magnitude varied from trial to trial in an unpredictable way (therefore could not be learned), but it is still possible that some participants may be trying to learn this way and therefore this model could perform well for those individuals.
- ExpLearn_decay includes the magnitude boosting mechanism as in our main ExpLearn model, but instead of a 50% fixed decay rate applied to unchosen and unrewarded options, this model estimates the decay rate as a fixed parameter.

We now report the results from these models’ fit in **Table S3**:

New Table S3, p.42:

Model	Description	N_{param}	AIC	OOS accuracy
ExpLearn_nomag	Magnitude is ignored, only learning of reward probability and outcomes are treated as binary (1: reward, 0: no reward)	2	212.3	0.524
ExpLearn_mag	Magnitude is incorporated in EV calculation and learned.	2	210.7	0.525
ExpLearn_decay	Magnitude is learned separately and boosts EV, decay rate (free parameter) applied to unchosen and unrewarded tokens.	4	208.7	0.539
ExpLearn	Model variant used in the main analyses reported in the manuscript. It is the same as ExpLearn_decay but with a fixed decay rate of 0.5.	3	207.7	0.539

Table S3. Summary of additional EL models tested during initial exploratory analyses of Study 1 data.

We found that the ExpLearn model performed best when accounting for model complexity, which is why we selected this model as our candidate model for the EL mechanism. We also found that the mean decay rate estimated from the ExpLearn_decay model was not different from 0.5 (mean=0.513, sd=0.227, $T(125)=0.639$, $P=0.52$), hence, providing a rationale for why we decided to use a fixed decay of 0.5 in our final EL model, thus minimizing model complexity.

As for the computation of OL and EL reliability, the choice of our computational approach is grounded in the previous literature – whereby using absolute prediction error as an index of uncertainty is justified from an algorithmic and neural implementation perspective. Prediction errors are known to be the key learning signal in reinforcement learning models, and absolute/unsigned prediction errors are commonly used to reflect surprise and are represented in the brain, making them a natural candidate to capture the (un)reliability of a learning strategy (O’Doherty et al, 2021). In terms of analyses supporting our implementation of reliability, we now provide extensive posterior predictive checks showing that the dynamic arbitration model can capture all expected aspects of behavior, and in particular the effects of uncertainty on each strategy, as seen in **Figure 4E-H** and in **Figure S5**. Two other analyses already present in the manuscript further speak to this as well: the finding that participants who rely primarily on the dynamic arbitration model exhibit the strongest behavioral index of arbitration (**Figure 7A-B**, formerly Figure 5A-B), and the finding that the trial-by-trial arbitration weight extracted from the model varies as expected according to high vs low uncertainty trials (**Figure S4**). We have added some content to further justify this in the Methods and Discussion that future work is needed to fully dissect whether other implementation of reliability could perform better in the context of arbitration, since this was outside the scope of the current study.

Methods, p.32: “Unsigned prediction errors were used as an index of how unreliable each strategy was, consistent with the hypothesis that when a strategy is reliable it should generate small prediction errors.”

Discussion, p.21: “Reliability in our model was defined using absolute prediction errors associated with each strategy as an index of uncertainty (or unreliability). This arbitration signal is consistent with the algorithmic and neural implementation of mixture of experts models in the literature (O’Doherty et al, 2021), though future work is needed to further explore whether other implementations of reliability could perform better. In particular, this could be achieved through more optimized task designs that fully allow distinguishing between different reliability computations, which was outside the scope of the current study.”

3.b. Regarding parameter recovery, in my understanding, averaging estimated parameters over 10 simulations before correlating them with the generative ones leads to grossly overestimating recovery, because this averages out the noise inherent in the estimation – hence does not give a fair estimate of the reliability of the parameters estimated in the actual dataset. A probably more fair approach would be to compute the correlation matrix in each of the 10 simulation independently, and then compute an averaged correlation matrix over simulations.

We have now performed the parameter recovery analysis in the way suggested by the reviewer: first, computing the correlation between actual and recovered parameters in each of the 10 simulations, then computing the average of these 10 correlations. The updated results are shown in **Figure S3B-F** (p.49). The correlations were indeed slightly weaker than with the previous analysis, but still showed, for all models and all parameters, high correlations between actual and recovered parameters ($R \sim 0.74$), higher than any existing cross-parameter correlations ($R < 0.36$).

4. Although I appreciate the attempt to tackle the heterogeneity of individual strategies, I have a feeling that in the present case, these results are more a symptom that the task (and/or models) does not capture a reliable process. In my understanding, the same results (all models seem represented in the data and there is no clear winner) would be obtained if no model is good at capturing any of the data (rather than the proposed interpretation that all models are somewhat good at capturing different share of the data). The “external validation” through the trans-psychiatric-like analysis is not overwhelmingly convincing (statistically speaking). Here, we come back to the necessity of obtaining clear and convincing assessment of the goodness-of-fit of the models (see my point 2).

We thank the reviewer for prompting us to further justify the validity of the heterogeneity approach, and believe that some of the analyses the reviewer suggested in their earlier comments have substantially helped strengthen that point. First, the out-of-sample accuracy analyses broken down by group (**Figure 5A-B**) show that within each group, we confirm out of sample that the predicted model performs best for that group. This is unsurprising, but if no model was good at capturing any of the data, we may have instead observed more confusion between models in that analysis. Second, we have performed posterior predictive checks broken down by group and corresponding model, for the learning curves (**Figure 5D-G**) and for the ME-GLM effects reflecting each strategy (**Figure 6A-D**). We have additionally repeated these two analyses by shuffling the group membership, that is by assigning each participant to one of the 5 groups at random (**Figure 5H-I** and **Figure 6E-F**). We see that when doing this, our ability to recover behavioral differences between the groups is substantially impaired, thus validating our heterogeneity approach and confirming that the models can each capture specific patterns in the data.

Results, p.13: “First, we found that calculating the five models’ out-of-sample accuracy separately for each group confirmed that each group was best fit by its respective model (Fig. 5A-B).”

Results, p.13: “Examining learning curves generated from each group’s best-fitting model (using individual participant’s best-fitting parameters within that group) showed an almost perfect match to the data (Fig. 5F-G compared to Fig. 5D-E, respectively), which was not observed when group membership was randomly shuffled (Fig. 5H-I).”

Results, p.15: “Additionally, posterior predictive checks confirmed that pattern of GLM effects between groups, whereby GLM effects from model-generated data matched the data well when split by actual groups (Fig. 6C-D), but not when split by randomly shuffled groups (Fig. 6E-F).”

5. Finally, I have some doubts about the conceptual validity of the experiment as an instance of “observational” learning. Arguably, the cover story of the task is pretty weak: the (fake) partner “making choices” presumably know the reward probability associated with each token; i.e. including the (un-signalized) reversals. This seems hardly believable, in which can wonder whether the task really investigates “observational” learning, or more credibly a hierarchical inference process akin to model-based/model-free types of dichotomies?

We thank the reviewer for raising this concern. First of all, we would like to clarify that we are not trying to claim that observational learning as a process is specifically “social”. When we use the term “observational”, we strongly believe that the inferences that we are modelling would be the same in the context of observing a computer agent rather than the choices of a (pretend) other player, which is why the believability of the cover story is not much of a concern here. Second, another important point that helps put this process in the category of “observational” learning, in ways that it cannot pertain to a model-based or model-free account of experiential learning, is that the observational learning part of the trial in this task can be learned solely from observing the choice of the other agent **in the absence of outcomes**, meaning that outcomes cannot be used to reinforce cached values (model-free) or build a model of the world (model-based). Finally, we do acknowledge that future work could extend the current findings to more naturalistic/ecologically valid ways of assessing observational learning, its modulation with the conditions of the environment and its arbitration with other learning strategies. We have added some content to the discussion related to these points.

Discussion, p.22: “This dichotomy goes beyond the well-known model-based/model-free arbitration^{20,21,53} given that the OL strategy is implemented in the absence of outcomes, that is by simply inferring the current goal from observing the other agent’s actions. In that sense, outcomes cannot be used to reinforce cached values (like in model-free learning or in the EL strategy here) or to build a model of the world (model-based learning).”

Discussion, p.23: “Another interesting question would also be to assess how the reliance on each strategy, and arbitration between them, may vary in response to more naturalistic and ecologically valid ways of implementing the strategies in the task.”

REVIEWERS' COMMENTS

Reviewer #1 (Remarks to the Author):

The authors have provided a thorough and compelling response and adequately addressed all my comments.

Reviewer #2 (Remarks to the Author):

I would like to thank the authors for their thorough consideration of our comments. I am satisfied with their responses, and have no further suggestions - I think this is a strong paper on a fascinating topic.

Of note - the point I was making about your mention of social skills/social learning in the introduction relates to your response to reviewer #3 - that OL does not only rely on social learning, but could also entail (though, admittedly, less commonly) learning from an agent or even from watching a replay of one's own actions a significant time later. This may be worth highlighting in the introduction where observational learning is introduced.

Reviewer #3 (Remarks to the Author):

In general, I commend the authors for a very serious and constructive revision. Overall, I find the manuscript much improved. There are still some framing and modelling choices that I find debatable, but they do not seem to jeopardize the main results nor the message of the paper. I just have a couple of minor concerns and suggestion that I hope the authors will find useful to polish their study.

1. I still find the depiction of the task somewhat confusing and deceptive. The Figure 1B should make clear that the variations of OL uncertainty correspond to independent blocks featuring new cues. I would also probably help the paper if the authors explicitly highlight on this panel the "reversal" points, which are then used extensively throughout the manuscript.

2. On a very personal note, I tend to find figures with overlapping data and model predictions more readable, and better at communicating what models capture and what they don't. I also don't think it is necessary to depict the full random effect in model predictions when it's not key to the figure. So, I encourage the authors to find a way to merge certain panels (couples of panels in Figure 4E-H; Figure 5D-F; Figure 5E-G; Figure 6A-C; Figure 6B-D). Shuffled model predictions could be in independent panels (or even in SI if needed).

February 20th, 2024

Dear Reviewers,

Thank you for reviewing the revised version of our manuscript “Heterogeneity in strategy use during arbitration between experiential and observational learning” (NCOMMS-23-16334-T). We have addressed the remaining comments from Reviewers #2 and #3 as follows.

Reviewer #2:

I would like to thank the authors for their thorough consideration of our comments. I am satisfied with their responses, and have no further suggestions - I think this is a strong paper on a fascinating topic.

Thank you!

Of note - the point I was making about your mention of social skills/social learning in the introduction relates to your response to reviewer #3 - that OL does not only rely on social learning, but could also entail (though, admittedly, less commonly) learning from an agent or even from watching a replay of one's own actions a significant time later. This may be worth highlighting in the introduction where observational learning is introduced.

As suggested, we have now highlighted in the introduction that Observational Learning does not only rely on specifically social/uniquely human processes, but extend to other domains:

(p.2, l. 14-15): *“OL is prevalent across many domains, [...] and can even extend to learning from non-human agents or from replayed actions.”*

Reviewer #3:

In general, I commend the authors for a very serious and constructive revision. Overall, I find the manuscript much improved. There are still some framing and modelling choices that I find debatable, but they do not seem to jeopardize the main results nor the message of the paper. I just have a couple of minor concerns and suggestion that I hope the authors will find useful to polish their study.

1. I still find the depiction of the task somewhat confusing and deceptive. The Figure 1B should make clear that the variations of OL uncertainty correspond to independent blocks featuring new cues. I would also probably help the paper if the authors explicitly highlight on this panel the “reversal” points, which are then used extensively throughout the manuscript.

Thank you for this suggestion. We have now edited Figure 1B to add depictions of the changes in blocks (shown as vertical dashed lines) and reversal points (shown as magenta triangles).

The new panel 1B is copied below for the reviewer’s reference:

2. On a very personal note, I tend to find figures with overlapping data and model predictions more readable, and better at communicating what models capture and what they don't. I also don't think it is necessary to depict the full random effect in model predictions when it's not key to the figure. So, I encourage the authors to find a way to merge certain panels (couples of panels in Figure 4E-H; Figure 5D-F; Figure 5E-G; Figure 6A-C; Figure 6B-D). Shuffled model predictions could be in independent panels (or even in SI if needed).

We have also merged some panels in Figure 4E-H, Figure 5D-F (now Figure 5C-D) and Figure 6 in order to show model predictions directly overlapping with the data.

- Figure 4E-H:

- Figure 5C-D:

- Figure 6:

Finally, we have moved the model predictions using simulated parameters and shuffled model predictions for the learning curves to the Supplementary Information, in Figure S8: